# STIEFEL FLOW MATCHING FOR MOMENT-CONSTRAINED STRUCTURE ELUCIDATION

**Austin H. Cheng**[1,2*] **Alston Lo**[1,2*] **Kin Long Kelvin Lee**[3] **Santiago Miret**[3]
**Alán Aspuru-Guzik**[1,2,4]
[1]University of Toronto  [2]Vector Institute  [3]Intel Labs  [4]Acceleration Consortium
https://github.com/aspuru-guzik-group/stiefelFM

## ABSTRACT

Molecular structure elucidation is a fundamental step in understanding chemical phenomena, with applications in identifying molecules in natural products, lab syntheses, forensic samples, and the interstellar medium. We consider the task of predicting a molecule's all-atom 3D structure given only its molecular formula and moments of inertia, motivated by the ability of rotational spectroscopy to measure these moments. While existing generative models can conditionally sample 3D structures with approximately correct moments, this soft conditioning fails to leverage the many digits of precision afforded by experimental rotational spectroscopy. To address this, we first show that the space of $n$-atom point clouds with a fixed set of moments of inertia is embedded in the Stiefel manifold $\mathrm{St}(n, 4)$. We then propose *Stiefel Flow Matching* as a generative model for elucidating 3D structure under exact moment constraints. Additionally, we learn simpler and shorter flows by finding approximate solutions for equivariant optimal transport on the Stiefel manifold. Empirically, enforcing exact moment constraints allows Stiefel Flow Matching to achieve higher success rates and faster sampling than Euclidean diffusion models, even on high-dimensional manifolds corresponding to large molecules in the GEOM dataset.

## 1 INTRODUCTION

Elucidating the structure of unknown molecules is a central task in chemistry, important for analyzing environmental samples (Moneta et al., 2023), identifying novel drugs (Sonstrom et al., 2023), and determining potential building blocks of life in the interstellar medium (McGuire et al., 2016). The challenge is to aggregate information from multiple sources of analytical data to unambiguously determine a molecule's structure. Rotational spectroscopy holds a unique capacity to provide precise measurements of a molecule's rotational constants, which are closely related to its moments of inertia. In turn, the connection between these moments and 3D structure has routinely provided the highest quality gas-phase 3D structures attainable from experiment (Domingos et al., 2020). Typically, structure elucidation with rotational spectroscopy proceeds by confirming whether a *known* structure's moments match with experiment (Lee & McCarthy, 2019; McCarthy et al., 2020). However, this approach is inherently restricted to molecules whose structures have already been catalogued, and leaves no prescription for *undiscovered* molecules such as novel natural products and key reactive intermediate species that cannot be easily isolated (Womack et al., 2015).

To overcome this limitation, we apply generative modeling to infer candidate 3D structures based on the moments and molecular formula *alone*. By themselves, moments of inertia are a 3-number summary of how a molecule's mass is distributed in space. Going from moments to 3D structure ($3 \to 3n$ values) is therefore a severely underconstrained inverse problem. Nevertheless, deep generative models such as diffusion (Ho et al., 2020; Song et al., 2020) and flow matching (Lipman et al., 2023; Liu et al., 2022) have shown promise in solving various inverse problems (Song et al., 2022; Chung et al., 2023; Song et al., 2023a). Indeed, previous work has applied Euclidean diffusion models to sample 3D structures conditioned on a given set of moments of inertia (Cheng et al., 2024).

---

*Equal contribution. <austin@cs.toronto.edu, alston.lo@mail.utoronto.ca>

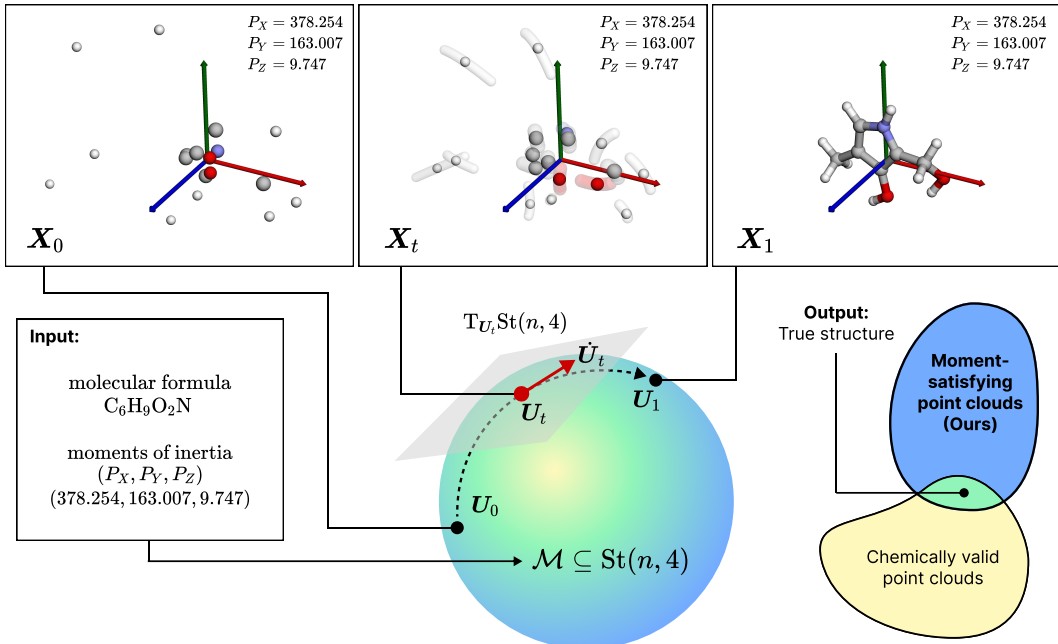

Figure 1: Stiefel Flow Matching learns to elucidate 3D molecular structure from moments and molecular formula alone by transforming uniform Stiefel noise $\boldsymbol{X}_0$ into *valid* molecular structures $\boldsymbol{X}_1$. Generative modelling on the Stiefel manifold $\mathrm{St}(n, 4)$ guarantees that samples always have the correct moments of inertia, which allows the network to focus only on generating chemically stable structures. Within the intersection of these spaces lies the true 3D structure.

However, analytic formulas for the moments of inertia are sufficiently simple that we can define the feasible space where these constraints are always exactly satisfied. Such precise adherence to moments can potentially leverage the many digits of precision provided by rotational spectroscopy (Shipman et al., 2011) in order to constrain the space of plausible structures. We first show that $\mathcal{M}$, the set of $n$-atom point clouds with fixed moments of inertia, is embedded in the Stiefel manifold $\mathrm{St}(n, 4)$. We then propose *Stiefel Flow Matching* as a generative model on the Stiefel manifold for solving the moment-constrained structure elucidation problem (Figure 1). Our approach augments Riemannian flow matching (Chen & Lipman, 2024) with equivariant optimal transport (Klein et al., 2023; Song et al., 2023c), which simplifies and shortens generation paths.

Concretely, our contributions are:

1. We propose the task of *moment-constrained structure elucidation* as a challenging generative modelling problem on the Stiefel manifold.

2. To solve this problem, we present *Stiefel Flow Matching*, a Riemannian flow matching approach. Furthermore, we formulate an objective for equivariant optimal transport on the Stiefel manifold, which obtains shorter and simpler flows.

3. Stiefel Flow Matching predicts 3D structure with greater success rate and lower cost than Euclidean diffusion models on both the QM9 (Ramakrishnan et al., 2014) and GEOM (Axelrod & Gomez-Bombarelli, 2022) datasets.

## 2   BACKGROUND AND APPROACH

We consider a 3D molecule as a point cloud of $n$ atoms with atomic numbers $\boldsymbol{a} \in \mathbb{N}^n$ and 3D coordinates $\boldsymbol{X} = (x_i, y_i, z_i)_{i=1}^N \in \mathbb{R}^{n \times 3}$. We also refer to $\boldsymbol{X}$ as the molecule's *3D structure*. Molecules have translational, rotational, and permutation symmetry. However, as $\boldsymbol{a}$ and $\boldsymbol{X}$ are stored on a computer, they necessarily have a node ordering and orientation. Therefore, when we refer to *structure*, we really mean the equivalence class containing $\boldsymbol{X}$ under these symmetries.

### 2.1 PROBLEM STATEMENT

We are given a molecule's molecular formula and moments of inertia, and wish to infer the 3D structure of the molecule. The molecular formula provides us with the number of atoms $n$, atomic numbers $\boldsymbol{a} \in \mathbb{N}^n$, and atomic masses $\boldsymbol{m} \in \mathbb{R}^n$. The moments of inertia[1] $(P_X, P_Y, P_Z)$ are three nonnegative numbers that summarize the mass distribution of the point cloud. They are defined as the eigenvalues of the *planar dyadic* $\boldsymbol{P} \in \mathbb{R}^{3 \times 3}$ (Kraitchman, 1953), calculated from masses $\boldsymbol{m}$ and the 3D atomic coordinates $\boldsymbol{X} = (x_i, y_i, z_i)_{i=1}^N \in \mathbb{R}^{n \times 3}$ as

$$\boldsymbol{P} = \boldsymbol{X}^\top (\operatorname{diag} \boldsymbol{m}) \boldsymbol{X} = \sum_{i=1}^n \begin{pmatrix} m_i x_i^2 & m_i x_i y_i & m_i x_i z_i \\ m_i y_i x_i & m_i y_i^2 & m_i y_i z_i \\ m_i z_i x_i & m_i z_i y_i & m_i z_i^2 \end{pmatrix}. \tag{1}$$

This definition assumes a coordinate system whose origin is the weighted center of mass of the molecule, which gives the constraint $\boldsymbol{m}^\top \boldsymbol{X} = \boldsymbol{0}$. We assume that $P_X > P_Y > P_Z > 0$, which eliminates rare edge cases (Appendix A.2). Note that $\boldsymbol{P}$ is symmetric and positive semidefinite and can be thought of as the mass-weighted covariance matrix of the point cloud, as in principal component analysis (PCA).

Diagonalizing $\boldsymbol{P}$ yields three eigenvectors, referred to as the *principal axes of rotation*, which orient a molecule in a canonical representation up to sign-flips. The principal axes correspond to directions which "explain" the most molecular mass, in analogy to PCA. The *principal axis system* is then the coordinate system whose origin is the center-of-mass and whose axes are the principal axes of rotation. We fix our coordinate system to be the principal axis system *by construction*, which gives the following constraints on $\boldsymbol{X}$:

$$\begin{aligned}
P_X &= \sum_{i=1}^n m_i x_i^2, & 0 &= \sum_{i=1}^n m_i y_i z_i, & 0 &= \sum_{i=1}^n m_i x_i, \\
P_Y &= \sum_{i=1}^n m_i y_i^2, & 0 &= \sum_{i=1}^n m_i x_i z_i, & 0 &= \sum_{i=1}^n m_i y_i, \\
P_Z &= \sum_{i=1}^n m_i z_i^2, & 0 &= \sum_{i=1}^n m_i x_i y_i, & 0 &= \sum_{i=1}^n m_i z_i.
\end{aligned} \tag{2}$$

These constraints also canonicalize a 3D structure up to sign-flips of the axes (e.g. $x \mapsto -x$). The center-of-mass constraint removes translational degrees of freedom, while the off-diagonal constraints remove rotational degrees of freedom.

The goal of molecular identification from moments and molecular formula is to find all molecular structures $\boldsymbol{X}$ which are consistent with these constraints *and* are thermodynamically *stable*, i.e., local minima of the potential energy surface. Structures which satisfy these criteria can then be compared to experimental measurements from rotational spectroscopy (Appendix A.3).

### 2.2 THE FEASIBLE SPACE OF MOMENT-CONSTRAINED STRUCTURES

The *Stiefel manifold* is the set of orthonormal $n \times p$ matrices, defined as

$$\operatorname{St}(n, p) = \{\boldsymbol{U} \in \mathbb{R}^{n \times p} \mid \boldsymbol{U}^\top \boldsymbol{U} = \boldsymbol{I}_p\}, \tag{3}$$

where $\boldsymbol{I}_p$ is the $p \times p$ identity matrix, and $n \geq p$. An element of $\operatorname{St}(n, p)$ can be thought of as the first $p$ columns of some $n$-dimensional (improper) rotation matrix, or can be thought of as a collection of $p$ orthonormal $n$-dimensional vectors. We provide more background in Appendix B.

Moment-constrained structures $\boldsymbol{X}$ can be mapped into $\operatorname{St}(n, 4)$ by scaling rows by masses and columns by moments. Letting $M = \sum_{i=1}^n m_i$ be the total mass, consider the following construction:

$$\boldsymbol{U} = \begin{pmatrix} \sqrt{\frac{m_1}{P_X}} x_1 & \sqrt{\frac{m_1}{P_Y}} y_1 & \sqrt{\frac{m_1}{P_Z}} z_1 & \sqrt{\frac{m_1}{M}} \\ \vdots & \vdots & \vdots & \vdots \\ \sqrt{\frac{m_n}{P_X}} x_n & \sqrt{\frac{m_n}{P_Y}} y_n & \sqrt{\frac{m_n}{P_Z}} z_n & \sqrt{\frac{m_n}{M}} \end{pmatrix} \in \mathbb{R}^{n \times 4}. \tag{4}$$

---

[1]This overview is a slight simplification of rotational spectroscopy. Full details are outlined in Appendix A.

It can be verified by inspection and comparison to the Equation 2 constraints that the columns of $U$ are orthonormal. That is, $U \in \mathrm{St}(n, 4)$. However, we cannot yet freely convert between $X$ and $U$: the last column of $U$ is not free as it must equal the unit mass vector $\hat{m} = \sqrt{m/M}$ to satisfy the zero center-of-mass constraint. To convert an arbitrary $U \in \mathrm{St}(n, 4)$ to an $X$ satisfying all constraints, we first apply a rigid $n$-dimensional rotation $R$ to $U$ so that its last column is aligned to $\hat{m}$ (Appendix B.5), before finally unscaling the rows and columns of $U$.

The feasible space $\mathcal{M}$ of moment-constrained structures is therefore the subset of $\mathrm{St}(n, 4)$ whose last column is fixed to $\hat{m}$, i.e.,

$$\mathcal{M} = \{U \in \mathrm{St}(n, 4) \mid U_{\cdot,4} = \hat{m}\}. \tag{5}$$

In fact, we show in Appendix B.7 that $\mathcal{M}$ is a totally geodesic submanifold of $\mathrm{St}(n, 4)$, which means that shortest paths between points in $\mathcal{M}$ stay in $\mathcal{M}$. The first three columns of elements in $\mathcal{M}$ form the intersection between $\mathrm{St}(n, 3)$ and the orthogonal complement to $\mathrm{span}(\hat{m})$, which is in turn equivalent to $\mathrm{St}(n - 1, 3)$. Hence, the dimension of $\mathcal{M}$ is $3n - 9$, which corresponds to removing 3 translational, 3 rotational, and 3 moment degrees of freedom, consistent with the 9 constraints in Equation 2. Going forward, we assume that the molecule of interest contains $n \geq 5$ atoms, so that we always deal with Stiefel manifolds of strictly rectangular matrices.

## 2.3 Navigating the Stiefel Manifold

The Stiefel manifold provides rich structure for navigating the feasible space of molecular structures. As a *manifold*, it is locally Euclidean but globally curved. This means that every point $U \in \mathrm{St}(n, p)$ is attached a vector space called its *tangent space* $\mathrm{T}_U \mathrm{St}(n, p)$. For the Stiefel manifold, these tangent spaces are given as

$$\mathrm{T}_U \mathrm{St}(n, p) = \{\Delta \in \mathbb{R}^{n \times p} \mid U^\top \Delta + \Delta^\top U = 0\}. \tag{6}$$

Then, equipping every tangent space with an inner product $\langle \cdot, \cdot \rangle_U$ turns $\mathrm{St}(n, p)$ into a *Riemannian manifold*, giving rise to notions of angles and distances. The collection of inner products for each tangent space is called the *Riemannian metric*. One such metric for the Stiefel manifold is the canonical metric (Edelman et al., 1998),

$$\langle \Delta, \tilde{\Delta} \rangle_U = \mathrm{trace}\, \Delta^\top \left(I_n - \tfrac{1}{2} U U^\top\right) \tilde{\Delta}, \tag{7}$$

which we exclusively use for this work. The canonical metric induces a norm $||\Delta||_U = \sqrt{\langle \Delta, \Delta \rangle_U}$ on each tangent space, which gives the length of a curve $\gamma \colon [0, 1] \to \mathrm{St}(n, p)$ as $L(\gamma) = \int_0^1 ||\dot{\gamma}(t)||_{\gamma(t)} dt$. Curves that are locally length-minimizing are called *geodesics*, providing a notion of "straight lines" for efficiently navigating around the manifold. Geodesics are defined by their starting point and initial velocity. Indeed, the exponential map $\exp_U(\Delta)$ takes in a starting point $U \in \mathrm{St}(n, p)$ and an initial velocity $\Delta \in \mathrm{T}_U \mathrm{St}(n, p)$, and outputs the final manifold point after following this geodesic for unit time. The exponential map is locally invertible, which gives the existence of the logarithmic map $\log_U(\tilde{U})$. The logarithmic map takes in a starting point $U$ and a target point $\tilde{U}$, and outputs the tangent vector needed to travel from $U$ to $\tilde{U}$. Algorithms for computing exponential and logarithmic maps under the canonical metric for the Stiefel manifold are given in Appendix B.3.

## 3 Stiefel Flow Matching

Having shown that the feasible space of moment-constrained structures is a Stiefel manifold, we can now formulate the problem of moment-constrained structure elucidation as an *unconstrained generative modeling problem on the Stiefel manifold*. An attractive approach to this is flow matching (FM), which trains a network as a time-dependent velocity field $u_t$ that transforms samples from a prior noise distribution $x_0 \sim p_0(x_0)$ into samples which approximately match the data distribution $x_1 \sim p_1(x_1) \approx p_{\mathrm{data}}(x_1)$ (Lipman et al., 2023). The integration of the velocity field over time is then a *continuous normalizing flow* $\psi_t$, which generates marginal probability densities $p_t$ over time by the pushforward operation $p_t = [\psi_t]_\# (p_0)$. In practice, this is realized by sampling initial conditions $x_0 \sim p_0(x_0)$ and evolving them from time 0 to $t$ according to the ODE $\frac{dx}{dt} = u_t(x)$ (Appendix C.4). The goal of training is to approximate this $u_t$ using a neural network $v_\theta(t, x)$ parameterized by $\theta$.

Flow matching is readily generalized to distributions on Riemannian manifolds (Chen & Lipman, 2024). When closed-form geodesics are available, Riemannian flow matching provides a simulation-

free training objective for learning $u_t$, called Riemannian conditional flow matching,

$$\mathcal{L}_{\text{RCFM}}(\theta) = \mathbb{E}_{t \sim U(0,1),\, p(\boldsymbol{U}_0),\, p(\boldsymbol{U}_1)} \left[ \|v_\theta(t, \boldsymbol{U}_t) - \dot{\boldsymbol{U}}_t\|^2_{\boldsymbol{U}_t} \right], \tag{8}$$

Intuitively, this loss trains the model to interpolate between many pairs of (noise, data). To compute this loss, we require (1) sampling $\boldsymbol{U}_0 \sim p(\boldsymbol{U}_0)$ from a prior noise distribution, (2) geodesic interpolation between $\boldsymbol{U}_0$ and $\boldsymbol{U}_1$ to get $\boldsymbol{U}_t$, (3) computing the time derivative of the interpolant $\dot{\boldsymbol{U}}_t$, and (4) evaluating the norm $\|\cdot\|_{\boldsymbol{U}_t}$.

To sample uniformly from the feasible space $\mathcal{M}$, we can sample $\boldsymbol{U}$ uniformly from $\text{St}(n, 4)$ (Appendix B.4) and then rigidly rotate $\boldsymbol{U}$ so that its last column aligns with the unit mass vector $\hat{\boldsymbol{m}}$ (Appendix B.5). Then, geodesics can be computed from the exponential (Edelman et al., 1998) and logarithmic (Zimmermann & Hüper, 2022) maps, which are efficient to compute for $\text{St}(n, 4)$,

$$\boldsymbol{U}_t = \exp_{\boldsymbol{U}_0}(t \log_{\boldsymbol{U}_0}(\boldsymbol{U}_1)), \tag{9}$$

making our training objective simulation-free. Once we have the interpolant $\boldsymbol{U}_t$, which will be the input to the neural network, we can then calculate the network's target $\dot{\boldsymbol{U}}_t$. Instead of autodifferentiation, which introduces unnecessary overhead, we compute $\dot{\boldsymbol{U}}_t$ using the logarithmic map,

$$\dot{\boldsymbol{U}}_t = \frac{1}{1-t} \log_{\boldsymbol{U}_t}(\boldsymbol{U}_1), \tag{10}$$

observing that $\dot{\boldsymbol{U}}_t$ is a unit length tangent vector along the geodesic from $\boldsymbol{U}_t$ to $\boldsymbol{U}_1$. Finally, we compute the norm of $v_\theta(t, \boldsymbol{U}_t) - \dot{\boldsymbol{U}}_t$ following Appendix B.2. Additionally, Appendix Theorem 4 shows how we can compute the logarithm in $\text{St}(n, 3)$, rather than $\text{St}(n, 4)$, which slightly saves time.

### 3.1 Reflection and Permutation Equivariance

As mentioned earlier, we set the coordinate axes as the principal axes of rotation by construction. This canonicalizes the 3D structure, removing translational symmetries and reducing the rotational symmetries to sign-flip symmetries of the eigenvectors of $\boldsymbol{P}$, which are the coordinate axes (Puny et al., 2022; Duval et al., 2023). Hence, the flow needs to be equivariant with respect to sign-flips of the coordinate axes, which we call "reflection-equivariance" for brevity (Lim et al., 2023; Cheng et al., 2024). In addition, because 3D structure is invariant under node order permutations, the learned velocity should be equivariant to permutations. Together, the explicit equivariance constraints on the network are given as $v_\theta(t, \boldsymbol{\Pi}\boldsymbol{U}_t\boldsymbol{R}) = \boldsymbol{\Pi}v_\theta(t, \boldsymbol{U}_t)\boldsymbol{R}$, for all node permutations $\boldsymbol{\Pi}$ and reflections $\boldsymbol{R} = \text{diag}(\pm 1, \pm 1, \pm 1, 1)$. We satisfy these constraints with a reflection-equivariant graph neural network architecture described in Appendix C.1.

### 3.2 Equivariant Optimal Transport

Flow matching learns a velocity field which transports samples from the noise distribution to the data distribution, but there is no guarantee that samples will follow paths $\gamma$ that are optimal with respect to transport cost $L(\gamma)$. Optimal paths are desired because they afford more efficient training and faster generation (Pooladian et al., 2023; Tong et al., 2024). Since molecules have permutation and rotational symmetries, optimal paths should connect structures whose equivalence classes are close to each other. This corresponds to finding optimal node permutations and rotations to align noise samples $\boldsymbol{X}_0$ to data samples $\boldsymbol{X}_1$. In contrast to the Euclidean case (Klein et al., 2023; Song et al., 2023c), only reflections are needed, not rotations, because the coordinate axes are already fixed in place as the principal axes of rotation by Equation 2.

In addition, transport cost on $\text{St}(n, 4)$ must be measured using the Riemannian distance. Thus, the optimal transport map from $\boldsymbol{U}_0$ to $\boldsymbol{U}_1$ minimizes the following cost over atom-type-preserving node permutations $\boldsymbol{\Pi}$ (i.e., $\boldsymbol{\Pi}\boldsymbol{a} = \boldsymbol{a}$) and reflections $\boldsymbol{R} = \text{diag}(\pm 1, \pm 1, \pm 1, 1)$:

$$d(\boldsymbol{U}_0, \boldsymbol{U}_1) = \|\log_{\boldsymbol{\Pi}\boldsymbol{U}_0\boldsymbol{R}}(\boldsymbol{U}_1)\|_{\boldsymbol{\Pi}\boldsymbol{U}_0\boldsymbol{R}}. \tag{11}$$

While searching for optimal alignments, we do not compute the full Stiefel logarithm and instead approximately calculate distance using only one iteration of the inner loop described in Appendix Algorithm 2, which we justify in Appendix E. This approximate distance is then heuristically optimized over atom permutations and reflections with a greedy random local search (Appendix D). Appendix Algorithm 4 outlines this procedure, which samples several permutations for each reflection to approximately identify the best reflection, and then refines the permutations by a local search of random atom type-preserving index swaps (Appendix Figure 9).

Table 1: Experimental results on QM9. Stiefel FM shows no violation of moment constraints as shown in the *Error* metrics, and has the highest success rate for structure elucidation, with the lowest computational cost.

| Method | % < RMSD ↑ | | Error ↓ | Valid ↑ | Stable ↓ | Diverse ↑ | NFE ↓ |
|---|---|---|---|---|---|---|---|
| | 0.25 Å | 0.10 Å | | | | | |
| Stiefel Random | 0.00±0.00 | 0.00±0.00 | 0.00 | 0.061 | nan | 2.640 | 0 |
| KREED | 11.22±0.28 | 9.55±0.26 | 5.18 | 0.878 | −1.335 | 1.429 | 1000 |
| KREED-XL | 13.65±0.30 | 10.94±0.27 | 3.64 | 0.933 | −1.048 | 0.870 | 1000 |
| KREED-XL-DPS | 12.36±0.29 | 9.40±0.26 | 1.33 | 0.744 | −0.826 | 1.060 | 1000 |
| KREED-XL-proj | 13.67±0.30 | 10.93±0.27 | 0.00 | 0.924 | −0.905 | 0.871 | 1000 |
| Stiefel FM | **15.17±0.31** | **13.82±0.30** | 0.00 | 0.882 | −1.125 | 1.040 | 200 |
| Stiefel FM-OT | 13.99±0.30 | 12.68±0.29 | 0.00 | 0.835 | −1.039 | 1.045 | 200 |

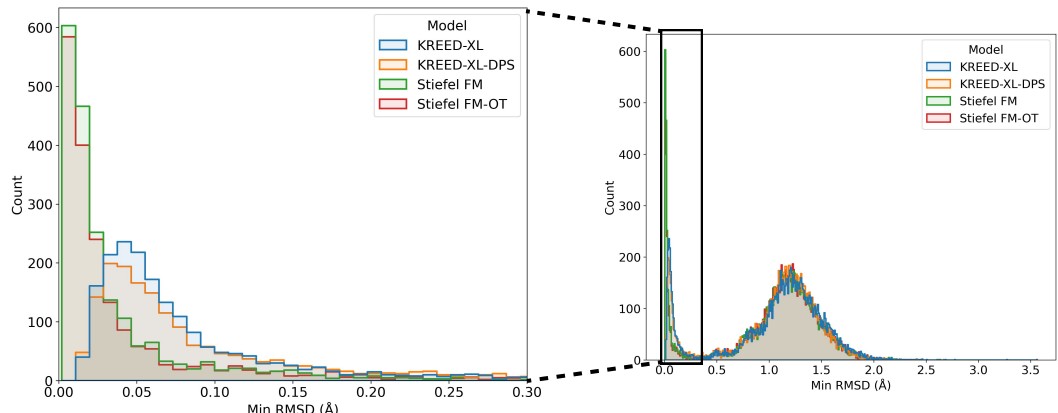

Figure 2: Histograms of minimum RMSD for predicted QM9 examples show two distinct clusters for RMSD. The 0.25Å threshold captures molecular structures that are useful for structure elucidation.

## 4 EXPERIMENTS

We evaluate Euclidean diffusion models and Stiefel Flow Matching on the QM9 and GEOM datasets. For each example, the model takes in moments and molecular formula and produces $K = 10$ samples.

**Datasets.** For QM9 (Ramakrishnan et al., 2014), we use the conformers provided by the GEOM dataset. We abbreviate GEOM-Drugs (Axelrod & Gomez-Bombarelli, 2022) as GEOM. We use the same training, validation, and test splits as Cheng et al. (2024), except we remove examples that are unstable, have less than 5 atoms, or have exactly zero-valued moments, which drops 3122 examples from QM9 and drops no examples from GEOM. QM9 has train/val/test splits of 104265/13056/13033 molecules, while GEOM has splits of 233625/29203/29203 molecules, or 5537598/29203/29203 conformers. We only predict 3D structures for the lowest-energy conformers of GEOM. This reflects experimental reality, as the lowest-energy conformer typically has the highest proportion in the population after cooling by supersonic jet expansion (Ruoff et al., 1990).

**Structure elucidation.** The only measure of a model's success is its ability to generate the correct 3D structure $X$ with high accuracy at least once, where accuracy is measured by root-mean-squared-deviation (RMSD) of the predicted coordinates to the ground truth. High accuracy is needed because the only confirming evidence available to us is (1) agreement with moments and (2) thermodynamic stability by quantum chemistry. For this reason, we use stringent thresholds of RMSD < 0.25 Å and RMSD < 0.10 Å to ensure that the predicted structure is in the same potential energy basin as the true structure. Success is reported if the minimum RMSD over $K = 10$ generated samples satisfies these thresholds. The success rate is the percentage of the test set whose generated samples has a *minimum* RMSD which satisfies each threshold. Error bars are standard errors of the mean.

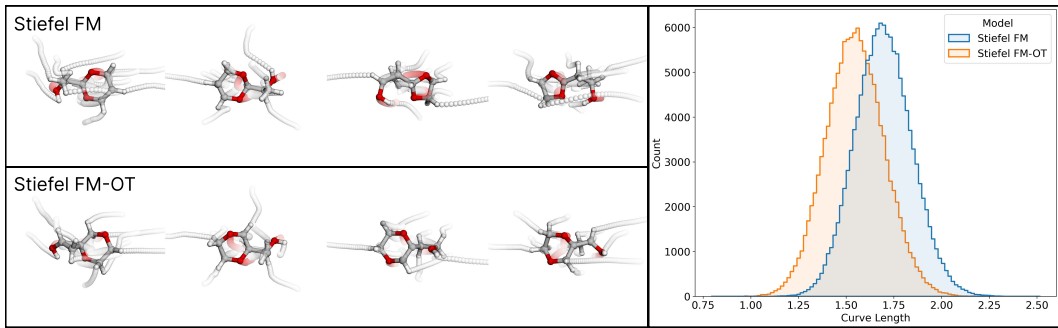

Figure 3: (*Left*) Learned sampling trajectories for Stiefel FM and Stiefel FM-OT on QM9. Each column begins generation from the same noise. (*Right*) Histogram of curve lengths of all QM9 sampling trajectories for Stiefel FM and Stiefel FM-OT. Permutation and reflection alignment lead to simpler and shorter paths.

Evaluating RMSD is nontrivial because the generated and ground truth structure must first be aligned under same-atom-type permutations and reflections. We use the same RMSD procedure as Cheng et al. (2024): We first align node permutations by solving a linear assignment problem whose cost matrix is squared Euclidean distance, and repeat this for all 8 reflections, taking the minimum. Then we compute RMSD between the ground truth coordinates and the aligned coordinates. This is similar to the alignment procedure used in (Klein et al., 2023; Song et al., 2023c).

**Auxiliary metrics.** To characterize the samples generated by each model, we report additional metrics. In contrast to success rate, which checks the minimum RMSD of generated samples for each example, these auxiliary metrics are averaged over *all* generated samples (except diversity). *Error* measures how much the generated structure violates the moment constraints. If $\hat{P}$ is the computed planar dyadic of the generated structure, this is computed as $\frac{1}{\sqrt{6}}||\text{triu}(\hat{P} - P)||_2$, where triu takes the upper triangular part of the matrix (contains 6 elements). *Validity* is a heuristic check based on bond detection with `rdDetermineBonds.DetermineConnectivity` (Landrum, 2013; Kim & Kim, 2015). *Stability* is the log norm of the gradient of energy with respect to coordinates, as reported by the `xtb` quantum chemistry program (Bannwarth et al., 2019). Stable structures should have a gradient norm close to zero (log norm very negative), assuming the structure is at a local minimum and not a saddle point. *Diversity* is calculated as the average pairwise RMSD of all generated samples of a single example. *NFE* is the number of function evaluations used during generation, and measures computational cost. We do not set boldface for these metrics because they do not correspond directly to success criteria.

**Baselines.** Given the novelty of the problem, the number of available baselines is limited. We compare the performance of Stiefel FM to KREED (Cheng et al., 2024). KREED is a reflection-equivariant diffusion model trained to generate 3D structure conditioned on molecular formula and moments of inertia, and is a specialization of E(3)-equivariant approaches like EDM (Hoogeboom et al., 2022). Since our architecture for Stiefel FM is much larger (parameter-wise) than the model architecture used in KREED, and because KREED is tailored for a slightly different task, we also train another reflection-equivariant diffusion model with an identical neural network architecture to Stiefel Flow Matching, which we label as KREED-XL. The planar dyadic is computable at every step of the generation process, which means that this task can be treated as a nonlinear inverse problem: On top of KREED-XL, we apply Diffusion Posterior Sampling (DPS) (Chung et al., 2023), which guides generation with an additional drift term for minimizing the planar dyadic error. As a simple baseline which exactly satisfies moment constraints, we report performance for uniform random sampling on the Stiefel manifold. We also report the results of KREED-XL after projecting samples onto the feasible manifold (Appendix B.6). Relevant hyperparameters for all methods are provided in Appendix C.3.

**Results.** Table 1 shows our experimental results on QM9, reporting Stiefel FM with and without optimal transport (OT). We note that the average number of atoms in QM9 is 18, meaning that on average the model must infer $3(18) - 9 = 39$ values from 3 moments. We find that Stiefel FM can generate the correct structure with a greater success rate than all Euclidean diffusion models. We

Table 2: Experimental results on GEOM. Stiefel FM generates few valid structures on its own due to the increased difficulty of manifold-constrained generative modelling. When adjusted to generate the same number of valid molecules, Stiefel FM-OT (filter) obtains the highest success rate, without surpassing baselines in computational cost.

| Method | % < RMSD ↑ | | Error ↓ | Valid ↑ | Stable ↓ | Diverse ↑ | NFE ↓ |
|---|---|---|---|---|---|---|---|
| | 0.25 Å | 0.10 Å | | | | | |
| Stiefel Random | 0.00±0.00 | 0.00±0.00 | 0.00 | 0.000 | nan | 4.104 | 0 |
| KREED | 0.04±0.01 | 0.02±0.01 | 58.36 | 0.353 | −0.583 | 2.286 | 1000 |
| KREED-XL | 3.54±0.11 | 2.02±0.08 | 30.71 | 0.907 | −0.900 | 2.190 | 1000 |
| KREED-XL-proj | 3.54±0.11 | 2.04±0.08 | 0.00 | 0.904 | −0.752 | 2.188 | 1000 |
| Stiefel FM | 2.17±0.09 | 1.24±0.06 | 0.00 | 0.388 | −0.066 | 2.212 | 200 |
| Stiefel FM-OT | 2.44±0.09 | 1.49±0.07 | 0.00 | 0.376 | −0.002 | 2.195 | 200 |
| Stiefel FM (filter) | 3.57±0.11 | 2.06±0.08 | 0.00 | 0.889 | −0.437 | 2.183 | 600 |
| Stiefel FM-OT (filter) | **3.94±0.11** | **2.42±0.09** | 0.00 | 0.869 | −0.352 | 2.165 | 600 |

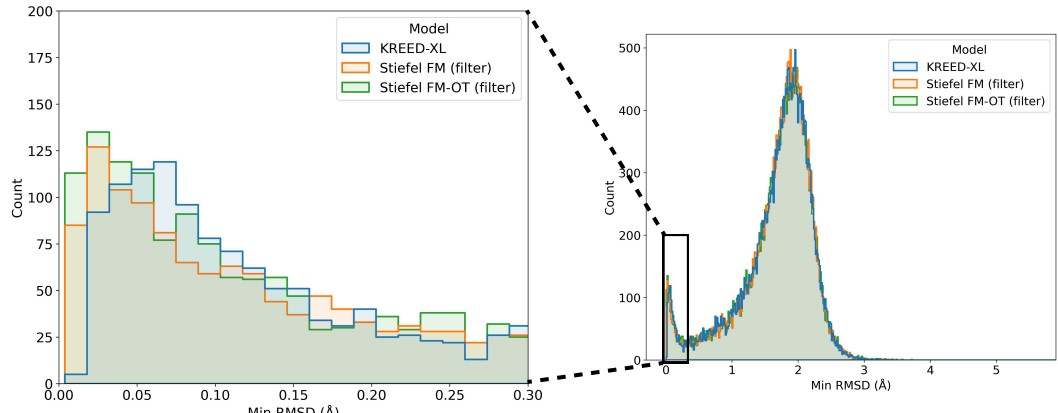

Figure 4: Histograms of minimum RMSD for predicted GEOM examples.

also see that incorporating the analytic formula of the moments via DPS does improve agreement with the moments, but at the cost of accuracy. In contrast, Stiefel FM does not suffer from this tradeoff. Projecting samples onto the manifold does not change success rate, because the projection leaves correct structures untouched, while only distorting incorrect structures. In addition, Stiefel FM uses only 20% of the computation used by diffusion models. Euclidean diffusion models such as KREED-XL can produce more valid and stable structures, though they may not necessarily generate the correct structure. Figure 2 reveals that when Stiefel FM's predictions are correct, it is likely to be extremely accurate, achieving RMSD even below 0.05 Å. Training with equivariant optimal transport helps learn simpler generation paths and reduces the average curve length of generation trajectories from 1.696 to 1.547 (Figure 3). Interestingly, training with optimal transport slightly reduces the success rate of Stiefel FM on QM9, though this trend is reversed for GEOM. Appendix Table 5 reports additional trials experimenting with optimal transport, logit-normal timestep sampling (Esser et al., 2024), and stochasticity (Bose et al., 2024).

The task of structure elucidation poses an even harder challenge on GEOM, with an average of 46 atoms, giving a task of inferring $3(46) − 9 = 129$ values. Naively, one would expect that this problem is hopelessly underconstrained. Nevertheless, Table 2 shows that both diffusion models and flow matching can obtain a nontrivial success rate for structure elucidation. However, the low accuracy, validity, and stability of Stiefel FM(-OT) suggests that the model is underfitting the dataset, even as KREED-XL is able to generate numerous valid samples. We observe that less than half of the structures generated by Stiefel FM are valid, which aligns with the fact that *there are fewer valid structures on the feasible manifold $\mathcal{M}$ than in regular Euclidean space*. This suggests that a fairer evaluation utilizing the strength of the manifold constraint should generate a similar number of

*valid* structures for each model. This is justified by the fact that validity does not use ground truth labels and can be computed at only nominal cost. Therefore, we generate 30 samples for Stiefel FM and Stiefel FM-OT before filtering to retain up to $K = 10$ valid samples. Note that the combined computational cost of generating 3x as many samples (3 x 200 NFE) is still lower than KREED-XL (1000 NFE). After filtering, Stiefel FM-OT obtains a similar validity rate to KREED-XL, but obtains the highest success rate for structure elucidation. Even after filtering, the mediocre stability of generated samples suggests that success rate can be improved further, though at the slightly higher cost of quantum chemistry calculations. Now, we see that optimal transport does help Stiefel FM to predict accurate structures, while also reducing average curve length from 1.421 to 1.344 (Appendix Figure 8). However, biasing generation by selecting for valid samples seems to also select for longer generation trajectories. We find that generation paths that land on the correct structure are usually longer than generation paths that land on incorrect structures (Appendix Figure 7). This may be explained by the fact that initial points are sampled anywhere uniformly on the manifold, but for success they must end up on the single true structure. In contrast, there are many incorrect structures all over the manifold, which may end up on average closer to random initial points.

One should expect success rates to be of this magnitude for solving a heavily underconstrained problem. This is simply due to the fact that there exist many stable structures with the same molecular formula and very similar moments of inertia. Indeed, trained models usually generate realistic-looking molecules – see generated examples in Appendix Figure 5 and Figure 6. Furthermore, only 10 to 30 samples were queried for each molecule, but an actual structure elucidation campaign would have a much larger compute budget for generating thousands of samples. As a highlight, our results show that it is actually possible at 0.25 Å resolution to elucidate 27.4% of the test set of QM9 (3580/13033) when combining KREED-XL, Stiefel FM, and Stiefel FM-OT; and 7.9% of the test set of GEOM (2297/29203), when combining KREED-XL, Stiefel FM (filter), and Stiefel FM-OT (filter).

## 5 RELATED WORK

**Generative models for 3D molecules.** While generative models have been actively explored for 3D structure prediction, few works have applied these methods to the task of structure elucidation (Cheng et al., 2024). Adjacent work in applying diffusion and flow matching models to 3D molecules include molecular generation (Hoogeboom et al., 2022; Song et al., 2023c), conformer search (Jing et al., 2022; Xu et al., 2022), docking (Corso et al., 2022), biomolecular assembly (Abramson et al., 2024) and Boltzmann generators (Klein et al., 2023). Generative models on Riemannian manifolds have also been applied to protein design (Bose et al., 2024; Yim et al., 2023) and crystal structure prediction (Jiao et al., 2024). Recent work has applied generative modelling to predict 3D molecular structure from powder X-ray diffraction patterns (Lai et al., 2024; Riesel et al., 2024) and 3D protein structure from cryo-EM density maps (Levy et al., 2024).

**Deep learning for molecular identification using rotational spectroscopy.** A limited number of works provide other parts of the complete workflow needed for structure elucidation using rotational spectroscopy. Zaleski & Prozument (2018) propose RAINet as a forward modeling approach for assigning rotational spectra, where a set of peaks is fed into a classifier, that categorizes the spectra to an appropriate multilayer perceptron that outputs spectroscopic parameters, including moments. McCarthy & Lee (2020) adopt a "mixture-of-experts" approach that maps spectroscopic parameters and approximate molecular formula into a set of complementary experimental observables and SMILES strings, though with limited success. Cheng et al. (2024) present a Euclidean diffusion model KREED for determining 3D structure from moments, molecular formula, and unsigned substitution coordinates, the latter of which is also measurable from rotational spectroscopy, but can be difficult and expensive to obtain. Most recently, Schwarting et al. (2024) provide a thorough analysis into the inverse problem, examining the frequency with which different molecules have moments of inertia that are very close in value. Stiefel Flow Matching could disambiguate these structures by providing a diversity of structures that satisfy moment constraints exactly.

**Statistics on the Stiefel manifold.** While the Stiefel manifold is often studied in the context of optimization (Absil et al., 2008; Chen et al., 2021; Kong et al., 2022), a number of works study probability distributions on the Stiefel manifold (Chakraborty & Vemuri, 2019; Chikuse, 1990). One distribution on the Stiefel manifold is called the *matrix von Mises-Fisher distribution* or *matrix Langevin distribution* (Pal et al., 2020; Chikuse, 2003; Jupp & Mardia, 1979). Wang & Solo (2020)

propose a particle filtering algorithm on the Stiefel manifold, with the first application of optimal transport on the Stiefel manifold. Yataka et al. (2023) propose a continuous normalizing flow on the Grassmann manifold, which is closely related to the Stiefel manifold.

# 6 CONCLUSION

We propose Stiefel Flow Matching, a Riemannian generative model for generating samples subject to exact orthogonality constraints, and apply it to the challenging inverse problem of structure elucidation from moments of inertia and molecular formula. Empirically, Stiefel Flow Matching achieves a higher success rate than Euclidean diffusion approaches. Satisfying the constraints exactly will enable future advances in Riemannian generative modelling to directly transfer to generating more stable molecules, without needing to consider agreement to the moments.

## 6.1 FUTURE RESEARCH DIRECTIONS

**Improving Stiefel generative models.** Riemannian flow matching empirically shows degradation compared to Riemannian diffusion (Lou et al., 2024; Zhu et al., 2024). This has been attributed to two pathologies of Riemannian flow matching for compact manifolds: (1) the geodesic-based velocity field is discontinuous at the cut locus (Lou et al., 2024; Zhu et al., 2024), and (2) the probability density has a shrinking support (Stark et al., 2024; Holderrieth et al., 2024). These pathologies may explain the difficulty of Stiefel FM in fitting GEOM, and motivate the development of alternative probability paths for Stiefel flow matching, such as Stiefel diffusion (De Bortoli et al., 2022), diffusion mixtures (Jo & Hwang, 2023), or flows which asymptotically land on the Stiefel manifold (Ablin & Peyré, 2022; Gao et al., 2022). Training Stiefel diffusion with the denoising score matching objective requires the heat kernel as training targets for the neural network, but these targets are expensive to compute (Azangulov et al., 2022; Lou et al., 2024). Alternatively, the matrix Langevin distribution may be amenable to modelling with Star-Shaped Denoising Diffusion Probabilistic Models (Okhotin et al., 2024). Future work can also explore varying the metric used on the Stiefel manifold, since the Stiefel manifold admits a 1-parameter family of metrics generalizing the canonical metric (Hüper et al., 2021). These metrics have efficient numerical algorithms for the exponential and logarithm (Mataigne et al., 2024). Sample-time advances in flow matching, such as corrector sampling (Gat et al., 2024) or enhancing the flow with a jump process (Holderrieth et al., 2024), are also orthogonal avenues for improvement.

A key limitation of Stiefel Flow Matching is the requirement of molecular formula as input. Recent approaches in discrete flow matching (Campbell et al., 2024; Gat et al., 2024) could enable a multimodal flow to simultaneously vary both continuous atom positions and discrete atom types. The jump processes of generator matching (Holderrieth et al., 2024) are particularly natural for this problem, as they could allow the flow to jump between Stiefel manifolds of different sizes.

**Modeling on real-world chemical data.** Since we require only minimal information in the moments and molecular formula, another direction is to incorporate other conditioning information, such as energy and force information, fragments of 2D graphs, dipole moments, or other sources of analytical chemistry data. We can do so through MCMC sampling (Du et al., 2023), guidance (Song et al., 2023a;b; Mardani et al., 2023), or diversity sampling approaches (Corso et al., 2024). This presents an opportunity in mass spectrometry, as the 3D structural information provided by the moments can distinguish molecules which have the same mass. Controllable diversity can also be leveraged here since, in a molecule identification campaign, we can eliminate candidates once experiment confirms that they are not correct. We then want to sample structures which are different from these eliminated candidates.

**Additional applications of Stiefel Flow Matching.** Stiefel generative models can also be applied to other domains of orthogonality constrained data, such as molecular orbitals (Mrovec & Berger, 2021; Aoto & da Silva, 2021), orthogonal neural network weights (Kong et al., 2022), and covariance matrices in neural data (Nejatbakhsh et al., 2024).

ACKNOWLEDGMENTS

We thank Luca Thiede, Kevin Xie, Adamo Young, and Karsten Kreis for helpful discussions. Molecular visualizations were created with `py3Dmol` (Rego & Koes, 2015). This research was undertaken thanks in part to funding provided to the University of Toronto's Acceleration Consortium from the Canada First Research Excellence Fund CFREF-2022-00042. Computational resources used in preparing this research were provided by the Acceleration Consortium. A.A.-G. thanks Anders G. Frøseth for his generous support. A.A.-G. also acknowledges the generous support of the Canada 150 Research Chairs program.

We acknowledge the Python community (Van Rossum et al., 1995; Oliphant, 2007) for developing the core set of tools that enabled this work, including PyTorch (Paszke et al., 2019), PyTorch Lightning (Falcon & The PyTorch Lightning team, 2019), PyTorch Geometric (Fey & Lenssen, 2019b), Geomstats (Miolane et al., 2020; 2024), einops (Rogozhnikov, 2022), RDKit (Landrum et al., 2024), py3Dmol (Rego & Koes, 2015), Jupyter (Kluyver et al., 2016), Matplotlib (Hunter, 2007), seaborn (Waskom, 2021), NumPy (Harris et al., 2020), SciPy (Virtanen et al., 2020), pandas (The pandas development team), pybind11 (W. Jakob, J. Rhinelander, D. Moldovan and others, 2017), and Eigen (Guennebaud, Jacob, et al., 2010).

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

## A ROTATIONAL SPECTROSCOPY & MOLECULAR STRUCTURE

### A.1 NOTATION

$(P_X, P_Y, P_Z)$ are really called the *planar moments of inertia*. We refer to them as the moments of inertia as shorthand. The actual *moments of inertia* $(I_A, I_B, I_C)$ are uniquely related to the planar moments by a simple linear transformation (Kraitchman, 1953).

### A.2 INERTIA EDGE CASES

We assume that the molecule of interest has moments of inertia $P_X > P_Y > P_Z > 0$, or in other words, is a nonplanar asymmetric rotor. This assumption holds for the vast majority of molecules. We now discuss edge cases, such as perfectly symmetric, planar, or linear molecules. It is worth noting that, owing to their rarity and symmetries, these edge cases have significant overlap with the set of molecules that have already been studied (McGuire, 2018).

When structures have two equal eigenvalues ($P_Y = P_Z$) in their inertia matrix, there is no longer a unique choice of these two principal axes. These axes now sweep out a plane of possibilities, and numerical diagonalization will arbitrarily pick two orthogonal axes from this plane. But, making an arbitrary choice does not break the mapping between $X$ and $U$ in Equation (4). The only issue is that to respect this additional symmetry, the flow should be invariant to in-plane rotoreflections of the molecule. However, these examples are so rare that they can be ignored: 76 examples in QM9 and 8 examples in GEOM. If needed, this symmetry can be handled using data augmentation.

Stiefel Flow Matching cannot handle *exactly* planar ($P_Z = 0$) and *exactly* linear ($P_Y = P_Z = 0$) molecules due to divide by zero in Equation (4). For this reason, 3109 examples are removed from QM9. The vast majority of "planar" molecules are actually slightly nonplanar and therefore pose no issue. In rare cases where molecules are truly planar, Stiefel Flow Matching can be reformulated for 1 and 2 dimensions.

### A.3 EXPERIMENTAL WORKFLOW

For an unknown molecule, we assume that its molecular formula can be measured by high-resolution mass spectrometry (Marshall & Hendrickson, 2008) and that its moments can be measured from rotational spectroscopy (Gordy et al., 1984). When not available, molecular formula may be guessed by brute force, informed by the size of the moments.

At a high level, rotational spectroscopy observes how molecules freely rotate in the gas phase. The rotation of molecules is *quantized*, giving rise to a discrete set of rotational states. Molecules can absorb or emit radiation at *characteristic* wavelengths to transition between these energy levels. Rotational spectroscopy measures the energies of these transitions. A broadband microwave spectrometer can simultaneously measure thousands of these transitions, producing a spectrum of many sharp peaks (Brown et al., 2006; 2008). Rotational transition energies are in the microwave to far infrared region, which is why rotational spectroscopy is also known as microwave spectroscopy.

For an asymmetric rigid molecule, and neglecting effects like centrifugal distortion and hyperfine structure, the molecule's rotational energy levels are essentially determined by three unique *rotational constants*, $A(BC), A > B > C$. Rotational constants are inversely proportional to the principal moments of inertia, i.e., $A = \frac{\hbar^2}{2I_A}$. Each energy level is also indexed by quantum numbers. Transitions are the differences in these energy levels. The molecule must have an appreciable dipole moment for these transitions to be measured.

Once a spectrum is measured, the rotational spectroscopist is tasked with assigning each transition to its quantum numbers and ultimately assigning rotational constants $A(BC)$. Spectral assignment is a challenging problem tackled in other works (Zaleski & Prozument, 2018; Yeh et al., 2019). Rotational constants can then simply be inverted to obtain effective moments of inertia.

Using these effective moments, a spectroscopist can now search for the true structure using any of the models developed in this work. In an actual structure elucidation campaign, only a few targets are considered, which permits querying $K > 1000$ samples for each target, and also leaves enough computational resources to evaluate the stability of every generated sample by quantum chemistry.

## A.4 EXPERIMENTAL PRECISION

While the proposed method generates structures which satisfy the moment constraints *exactly*, and while rotational spectroscopy can measure experimental rotational constants to many digits of precision (Vogt et al., 2011), it is unfortunate that the experimental rotational constants do not directly translate to moments of inertia. This is because molecules are not perfectly rigid: Experiment observes properties that have been *vibrationally* averaged, including the rotational constants. Conformational fluctuations such as torsions can be frozen out by cooling molecules to their ground vibrational state. However, even in the ground vibrational state, a molecule is still vibrating due to zero-point energy. As a result, the experimental rotational constants $A(BC)_0$ are proportional to $\langle \frac{1}{r^2} \rangle$, whereas equilibrium rotational constants $A(BC)_e$ are proportional to $\frac{1}{\langle r \rangle^2}$, where $\langle \cdot \rangle$ denotes a vibrational average. Structures in QM9 and GEOM have been geometry optimized to reach equilibrium structures $\langle r \rangle$. This error due to zero-point vibration effects is the major source of uncertainty between equilibrium and experimental rotational constants, on the order of 1% relative error (Vogt et al., 2011; Puzzarini & Stanton, 2023).

Typically, experimental rotational constants $A(BC)_0$ can be corrected into equilibrium rotational constants $A(BC)_e$ by a rovibrational calculation. Experimental and equilibrium rotational constants are related by a perturbative expansion (Demaison et al., 2011)

$$A(BC)_0 = A(BC)_e - \frac{1}{2} \sum_i \alpha_i^{A(BC)} + \frac{1}{8} \sum_{ij} \gamma_{ij}^{A(BC)} + \cdots, \tag{12}$$

where $i, j$ index normal modes of vibration ($i, j \in \{1, 2, \ldots, 3n - 6\}$ for $n$ atoms), and $\alpha$ and $\gamma$ are first and second order interaction constants that couple rotational and vibrational motions together. As a Taylor series, $\alpha$ is generally much larger than $\gamma$, so $\gamma$ is usually neglected. The rovibrational correction $\alpha$ can be calculated using an electronic structure method with a good compromise between computational time and accuracy (Puzzarini & Stanton, 2023; Spaniol et al., 2023). However, this calculation requires knowing the structure in the first place.

Alternatively, experimental rotational constants can be approximately corrected to equilibrium rotational constants by simple empirical scaling factors (Lee & McCarthy, 2020).

But, exact moment constraints do allow application in the following sense: if one were to *a priori* guess the equilibrium moments correctly, one could then verify whether they are indeed correct by generating structures, calculating their rovibrational corrections, and then checking their agreement to the experimental rotational constants. Therefore, given experimental rotational constants $A(BC)_0$, one can use this verification procedure in a fine-grid search for the true equilibrium rotational constants $A(BC)_e$. This is feasible since $A(BC)_e$ consist of only 3 numbers. Experimental precision in $A(BC)_0$ is maintained up to the precision in computing $\alpha$. To verify that a structure is the true structure, we must know that (1) its $A(BC)_e$ and $\alpha$ match $A(BC)_0$ and (2) its gradient norm is 0.

## B THE STIEFEL MANIFOLD

In this section, we discuss various facts about the Stiefel manifold and computations thereon. We refer readers to (Lee, 2003; Edelman et al., 1998; Bendokat et al., 2024) for further details.

The Stiefel manifold $\text{St}(n, p)$ is the set of rectangular orthonormal matrices of shape $n \times p$:

$$\text{St}(n, p) = \{ \boldsymbol{U} \in \mathbb{R}^{n \times p} \mid \boldsymbol{U}^\top \boldsymbol{U} = \boldsymbol{I}_p \}. \tag{13}$$

$\text{St}(n, p)$ is a manifold of dimension $np - \frac{1}{2}p(p + 1)$. An element of $\text{St}(n, p)$ can be thought of as a collection of $p$ orthonormal $n$-dimensional vectors (i.e. a $p$-frame living in $n$-dimensional space), or as the first $p$ columns of an $n \times n$ orthogonal matrix. In fact, $\text{St}(n, p)$ generalizes some well-known spaces. For example, $\text{St}(n, n)$ is the orthogonal group $\text{O}(n)$, and $\text{St}(n, n - 1)$ is diffeomorphic to the special orthogonal group $\text{SO}(n)$, and $\text{St}(n, 1)$ recovers the unit $n$-sphere $S^{n-1}$.

### B.1 TANGENT SPACE

The tangent space of $\text{St}(n, p)$ at a point $\boldsymbol{U}$ is identified with the subspace

$$\text{T}_{\boldsymbol{U}} \text{St}(n, p) = \{ \Delta \in \mathbb{R}^{n \times p} \mid \boldsymbol{U}^\top \Delta + \Delta^\top \boldsymbol{U} = \boldsymbol{0} \}. \tag{14}$$

In other words, this is the set of $n \times p$ matrices $\Delta$ for which $\boldsymbol{U}^\top \Delta$ is skew-symmetric.

## B.2 Canonical metric

Smoothly equipping every tangent space with an inner product turns $\text{St}(n, p)$ into a Riemannian manifold. We exclusively consider the canonical metric,

$$\langle \Delta, \tilde{\Delta} \rangle_{\boldsymbol{U}} = \text{trace } \Delta^\top \left( \boldsymbol{I}_n - \tfrac{1}{2} \boldsymbol{U} \boldsymbol{U}^\top \right) \tilde{\Delta}. \tag{15}$$

Any inner product induces a norm by $||\Delta||_{\boldsymbol{U}} = \sqrt{\langle \Delta, \Delta \rangle_{\boldsymbol{U}}}$. For the canonical metric, the squared norm can be rearranged as:

$$||\Delta||^2_{\boldsymbol{U}} = \langle \Delta, \Delta \rangle_{\boldsymbol{U}} = \text{trace } \Delta^\top \Delta - \tfrac{1}{2} \text{trace}(\boldsymbol{U}^\top \Delta)^\top (\boldsymbol{U}^\top \Delta) = ||\Delta||^2_F - \tfrac{1}{2} ||\boldsymbol{U}^\top \Delta||^2_F, \tag{16}$$

where $|| \cdot ||_F$ is the Frobenius norm. Computing the canonical norm in this way is much faster for the node-wise graph batching approach used by PyTorch Geometric (Fey & Lenssen, 2019a).

## B.3 Exponential and logarithm

The Stiefel exponential can be computed by the algorithm presented by Edelman et al. (1998), who give the closed-form expression for a geodesic $\gamma$ with initial conditions $\gamma(0) = \boldsymbol{U}$ and $\dot{\gamma}(0) = \Delta$. Here, we reproduce the algorithm for computing $\exp_{\boldsymbol{U}}(\Delta) = \gamma(1)$.

---

**Algorithm 1** Computing a Stiefel geodesic $\gamma(t)$. (Edelman et al., 1998)

---

**Require:** Base point $\boldsymbol{U} \in \text{St}(n, p)$, initial direction $\Delta \in \text{T}_{\boldsymbol{U}}\text{St}(n, p)$, time $t \in \mathbb{R}$
1: $\boldsymbol{Q}\boldsymbol{R} \leftarrow (\boldsymbol{I}_n - \boldsymbol{U}\boldsymbol{U}^\top)\Delta$, where $\boldsymbol{Q} \in \mathbb{R}^{n \times p}$ and $\boldsymbol{R} \in \mathbb{R}^{p \times p}$        ▷ QR decomposition
2: $\boldsymbol{A} \leftarrow \begin{pmatrix} \boldsymbol{U}^\top \Delta & -\boldsymbol{R}^\top \\ \boldsymbol{R} & \boldsymbol{0} \end{pmatrix} \in \mathbb{R}^{2p \times 2p}$
3: $\left( \begin{array}{c|c} \boldsymbol{M}(t) & \cdots \\ \hline \boldsymbol{N}(t) & \cdots \end{array} \right) \leftarrow \exp_m(t\boldsymbol{A})$, where $\boldsymbol{M}(t), \boldsymbol{N}(t) \in \mathbb{R}^{p \times p}$        ▷ take submatrices
4: **return** $\boldsymbol{U}\boldsymbol{M}(t) + \boldsymbol{Q}\boldsymbol{N}(t) \in \text{St}(n, p)$

---

An efficient algorithm for computing the Stiefel logarithm is reproduced and simplified here from Zimmermann & Hüper (2022). We implement the logarithm in C++ for speed on the CPU, as it must be called on every fetch of a data example. This does not present a bottleneck: computing logarithmic maps takes an average of 0.1 ms for molecules in both QM9 and GEOM. The logarithm has a cost of $O(np^2)$. The thin QR decomposition and initial matrix multiplications have a cost of $O(np^2)$, while the inner loop's Schur decomposition, Sylvester solve, matrix multiplications, and matrix exponential have a cost of $O(p^3)$. We only execute the inner loop a maximum of 20 times (see Appendix E for convergence).

---

**Algorithm 2** Computing the Stiefel logarithm. (Zimmermann & Hüper, 2022)

---

**Require:** Base point $\boldsymbol{U} \in \text{St}(n, p)$, target point $\tilde{\boldsymbol{U}} \in \text{St}(n, p)$
1: $\boldsymbol{M} \leftarrow \boldsymbol{U}^\top \tilde{\boldsymbol{U}} \in \mathbb{R}^{p \times p}$
2: $\boldsymbol{Q}\boldsymbol{N} \leftarrow \tilde{\boldsymbol{U}} - \boldsymbol{U}\boldsymbol{M} \in \mathbb{R}^{n \times p}$        ▷ thin QR
3: $\boldsymbol{O}\boldsymbol{R} \leftarrow \begin{pmatrix} \boldsymbol{M} \\ \boldsymbol{N} \end{pmatrix} \in \mathbb{R}^{2p \times p}, \boldsymbol{O} \in \mathbb{R}^{2p \times 2p}$        ▷ compute orthogonal completion via full QR
4: $\boldsymbol{V} \leftarrow \begin{pmatrix} \boldsymbol{M} & \boldsymbol{O}_{\cdot, p:2p} \\ \boldsymbol{N} & \end{pmatrix} \in \text{SO}(2p)$        ▷ flip sign if $\det < 0$
5: **for** $k = 1, \ldots, 20$ **do**
6:     $\begin{pmatrix} \boldsymbol{A} & -\boldsymbol{B}^\top \\ \boldsymbol{B} & \boldsymbol{C} \end{pmatrix} \leftarrow \log_m(\boldsymbol{V})$        ▷ use Schur matrix logarithm
7:     $\boldsymbol{S} \leftarrow \tfrac{1}{12} \boldsymbol{B}\boldsymbol{B}^\top - \tfrac{1}{2} \boldsymbol{I}_p$
8:     solve $\boldsymbol{C} = \boldsymbol{S}\boldsymbol{\Gamma} + \boldsymbol{\Gamma}\boldsymbol{S}$ for $\boldsymbol{\Gamma} \in \mathbb{R}^{p \times p}$        ▷ symmetric Sylvester equation
9:     $\boldsymbol{\Phi} \leftarrow \exp_m(\boldsymbol{\Gamma}) \in \mathbb{R}^{p \times p}$        ▷ matrix exponential, or use Cayley transform
10:     $\boldsymbol{V}_{\cdot, p:2p} \leftarrow \boldsymbol{V}_{\cdot, p:2p} \boldsymbol{\Phi}$        ▷ rotate last $p$ columns of $\boldsymbol{V}$
11: **return** $\boldsymbol{U}\boldsymbol{A} + \boldsymbol{Q}\boldsymbol{B} \in \text{T}_{\boldsymbol{U}}\text{St}(n, p)$

---

## B.4 Uniform sampling

To sample uniformly on $\mathrm{St}(n,p)$ with respect to the Haar measure, we can compute $\boldsymbol{U} = \boldsymbol{Z}(\boldsymbol{Z}^\top \boldsymbol{Z})^{-1/2}$ for a random matrix $\boldsymbol{Z} \in \mathbb{R}^{n \times p}$ whose elements are drawn i.i.d. from a standard Gaussian (i.e. $\boldsymbol{Z} = \texttt{randn(n, p)}$).

## B.5 Householder reflections

To align the final column of a matrix $\boldsymbol{U} \in \mathrm{St}(n,p)$ to a unit vector $\boldsymbol{y}$, we can rotate its columns under a transformation $\boldsymbol{R} \in \mathrm{SO}(n)$ such that $\boldsymbol{R}\boldsymbol{U}_{.,4} = \boldsymbol{y}$. Let $\boldsymbol{U}_{.,4} = \boldsymbol{x}$. For any unit vector $\boldsymbol{v} \in S^{n-1}$, the Householder matrix $\boldsymbol{H}(\boldsymbol{v}) = \boldsymbol{I}_n - 2\boldsymbol{v}\boldsymbol{v}^\top$ has determinant $-1$. Then,

$$\boldsymbol{R} = \boldsymbol{H}(\boldsymbol{y})\boldsymbol{H}\left(\frac{\boldsymbol{x}+\boldsymbol{y}}{||\boldsymbol{x}+\boldsymbol{y}||}\right) \in \mathrm{SO}(n) \tag{17}$$

is our desired rotation. In particular, we can verify that

$$\boldsymbol{H}\left(\frac{\boldsymbol{x}+\boldsymbol{y}}{||\boldsymbol{x}+\boldsymbol{y}||}\right)\boldsymbol{x} = \boldsymbol{x} - \frac{(2+2\boldsymbol{x}^\top\boldsymbol{y})}{||\boldsymbol{x}+\boldsymbol{y}||^2}(\boldsymbol{x}+\boldsymbol{y}) = \boldsymbol{x} - (\boldsymbol{x}+\boldsymbol{y}) = -\boldsymbol{y} \tag{18}$$

so that $\boldsymbol{R}\boldsymbol{x} = \boldsymbol{H}(\boldsymbol{y})(-\boldsymbol{y}) = \boldsymbol{y}$, as desired.

## B.6 Orthogonal projections

The orthogonal projection of a point $\boldsymbol{U}_0 \in \mathbb{R}^{n \times p}$ onto $\mathrm{St}(n,p)$ is a special case of the well-known orthogonal Procrustes problem:

$$\arg\min_{\boldsymbol{U}} ||\boldsymbol{U}\boldsymbol{I}_p - \boldsymbol{U}_0||_F, \quad \text{subject to } \boldsymbol{U}^\top\boldsymbol{U} = \boldsymbol{I}_p. \tag{19}$$

Letting $\boldsymbol{U}_0 = \boldsymbol{A}\boldsymbol{\Sigma}\boldsymbol{B}^\top$ under a singular value decomposition, the solution to this problem is $\boldsymbol{A}\boldsymbol{B}^\top$. The orthogonal projection of a point $\boldsymbol{Z} \in \mathbb{R}^{n \times p}$ onto the tangent space $\mathrm{T}_{\boldsymbol{U}}\mathrm{St}(n,p)$ is computed by $\boldsymbol{Z} - \boldsymbol{U}\,\mathrm{sym}(\boldsymbol{U}^\top\boldsymbol{Z})$, where $\mathrm{sym}(\boldsymbol{A}) = \frac{1}{2}(\boldsymbol{A}+\boldsymbol{A}^\top)$ is a symmetrization operation.

## B.7 The zero center-of-mass submanifold

For $n \geq 5$, we are interested in the subset of $\mathrm{St}(n,4)$ obtained by fixing the final column to a fixed unit vector $\boldsymbol{a} \in S^{n-1}$:

$$\mathcal{M} = \{\boldsymbol{U} \in \mathrm{St}(n,p) \mid \boldsymbol{U}_{.,4} = \boldsymbol{a}\}. \tag{20}$$

Note that $S = \pi^{-1}(\{\boldsymbol{a}\})$ is a level set of the projection map

$$\pi\colon \mathrm{St}(n,4) \to S^{n-1}, \quad \boldsymbol{U} \mapsto \boldsymbol{U}_{.,4}. \tag{21}$$

In fact, $\pi$ is a smooth surjective map of constant rank, so it is a submersion by the global rank theorem. Hence, $\mathcal{M}$ is an embedded submanifold by the submersion level set theorem. The tangent vectors to $\mathcal{M}$ are exactly those whose final column is zero, since

$$\mathrm{T}_{\boldsymbol{U}}\mathcal{M} = \ker \mathrm{d}\pi_{\boldsymbol{U}} = \{\Delta \in \mathrm{T}_{\boldsymbol{U}}\mathrm{St}(n,4) \mid \Delta_{.,4} = \boldsymbol{0}\}. \tag{22}$$

If $\mathcal{M}$ further inherits the canonical metric from $\mathrm{St}(n,4)$, then it becomes a Riemannian manifold. Note that $\mathcal{M}$ is homeomorphic to $\mathrm{St}(n-1,3)$, so it is connected and compact. Hence, it is geodesically complete and any two points in $\mathcal{M}$ can be connected with a length-minimizing geodesic on $\mathcal{M}$. Theorem 1 shows that $\mathcal{M}$ is totally geodesic, i.e., any geodesic on $\mathcal{M}$ is a geodesic on $\mathrm{St}(n,4)$. This allows us to perform simpler computations in the ambient space. Theorem 5 computes projections of arbitrary matrices onto $\mathrm{T}_{\boldsymbol{U}}\mathcal{M}$.

**Theorem 1.** *As defined above, $\mathcal{M}$ is totally geodesic.*

*Proof.* A sufficient condition is that for any $\boldsymbol{U} \in \mathcal{M}$ and $\Delta \in \mathrm{T}_{\boldsymbol{U}}\mathcal{M}$, the geodesic $\gamma$ with initial conditions $\gamma(0) = \boldsymbol{U}$ and $\dot{\gamma}(0) = \Delta$ stays within $\mathcal{M}$. Fortunately, $\gamma$ is given in closed-form by Edelman et al. (1998) in Algorithm 1. Since $\Delta_{.,4} = \boldsymbol{0}$, the fourth columns of $\boldsymbol{R}$ and $\boldsymbol{U}^\top\Delta$ are zero. Since $\boldsymbol{U}^\top\Delta$ is skew-symmetric by Equation 14, its fourth row is also zero. Then, the fourth row and

column of $\boldsymbol{A}$ and $\exp_m(t\boldsymbol{A})$ are zero, but $\exp_m(t\boldsymbol{A})_{4,4} = 1$. It follows that $\boldsymbol{M}(t)_{\cdot,4} = (0,0,0,1)$ and $\boldsymbol{N}(t)_{\cdot,4} = \boldsymbol{0}$, so that

$$\gamma(t)_{\cdot,4} = \boldsymbol{U}\left(\boldsymbol{M}(t)_{\cdot,4}\right) + \boldsymbol{Q}\left(\boldsymbol{N}(t)_{\cdot,4}\right) = \boldsymbol{U}_{\cdot,4} = \boldsymbol{a}. \tag{23}$$

Hence, $\gamma(t) \in \mathcal{M}$ as desired. $\square$

We now provide some lemmas which are useful for proving that the Stiefel logarithm on $\mathcal{M}$ can be computed using $\mathrm{St}(n,3)$ rather than $\mathrm{St}(n,4)$ (Theorem 4).

**Lemma 2.** *Let $\boldsymbol{X}$ be a square matrix whose $i^{th}$ row and column are 0. Positive powers of $\boldsymbol{X}$ retain zeros in the $i^{th}$ row and column. Specifically, suppose*

$$\boldsymbol{X} = \begin{pmatrix} \boldsymbol{E} & \boldsymbol{0} & \boldsymbol{F} \\ \boldsymbol{0} & 0 & \boldsymbol{0} \\ \boldsymbol{G} & \boldsymbol{0} & \boldsymbol{H} \end{pmatrix}, \qquad \begin{pmatrix} \boldsymbol{E} & \boldsymbol{F} \\ \boldsymbol{G} & \boldsymbol{H} \end{pmatrix}^k = \begin{pmatrix} \hat{\boldsymbol{E}} & \hat{\boldsymbol{F}} \\ \hat{\boldsymbol{G}} & \hat{\boldsymbol{H}} \end{pmatrix}. \tag{24}$$

*Then we have*

$$\boldsymbol{X}^k = \begin{pmatrix} \hat{\boldsymbol{E}} & \boldsymbol{0} & \hat{\boldsymbol{F}} \\ \boldsymbol{0} & 0 & \boldsymbol{0} \\ \hat{\boldsymbol{G}} & \boldsymbol{0} & \hat{\boldsymbol{H}} \end{pmatrix}. \tag{25}$$

*Proof.* We show this by induction over $k$. It is clear this is true for $k = 1$. Then, we assume $\boldsymbol{X}^k = \begin{pmatrix} \hat{\boldsymbol{E}} & \boldsymbol{0} & \hat{\boldsymbol{F}} \\ \boldsymbol{0} & 0 & \boldsymbol{0} \\ \hat{\boldsymbol{G}} & \boldsymbol{0} & \hat{\boldsymbol{H}} \end{pmatrix}$ and aim to show for $k+1$. We know that

$$\begin{pmatrix} \boldsymbol{E} & \boldsymbol{F} \\ \boldsymbol{G} & \boldsymbol{H} \end{pmatrix}^{k+1} = \begin{pmatrix} \hat{\boldsymbol{E}} & \hat{\boldsymbol{F}} \\ \hat{\boldsymbol{G}} & \hat{\boldsymbol{H}} \end{pmatrix}\begin{pmatrix} \boldsymbol{E} & \boldsymbol{F} \\ \boldsymbol{G} & \boldsymbol{H} \end{pmatrix} = \begin{pmatrix} \hat{\boldsymbol{E}}\boldsymbol{E} + \hat{\boldsymbol{F}}\boldsymbol{G} & \hat{\boldsymbol{E}}\boldsymbol{F} + \hat{\boldsymbol{F}}\boldsymbol{H} \\ \hat{\boldsymbol{G}}\boldsymbol{E} + \hat{\boldsymbol{H}}\boldsymbol{G} & \hat{\boldsymbol{G}}\boldsymbol{F} + \hat{\boldsymbol{H}}\boldsymbol{H} \end{pmatrix}. \tag{26}$$

At the same time, we have that

$$\boldsymbol{X}^{k+1} = \begin{pmatrix} \hat{\boldsymbol{E}} & \boldsymbol{0} & \hat{\boldsymbol{F}} \\ \boldsymbol{0} & 0 & \boldsymbol{0} \\ \hat{\boldsymbol{G}} & \boldsymbol{0} & \hat{\boldsymbol{H}} \end{pmatrix}\begin{pmatrix} \boldsymbol{E} & \boldsymbol{0} & \boldsymbol{F} \\ \boldsymbol{0} & 0 & \boldsymbol{0} \\ \boldsymbol{G} & \boldsymbol{0} & \boldsymbol{H} \end{pmatrix} = \begin{pmatrix} \hat{\boldsymbol{E}}\boldsymbol{E} + \hat{\boldsymbol{F}}\boldsymbol{G} & \boldsymbol{0} & \hat{\boldsymbol{E}}\boldsymbol{F} + \hat{\boldsymbol{F}}\boldsymbol{H} \\ \boldsymbol{0} & 0 & \boldsymbol{0} \\ \hat{\boldsymbol{G}}\boldsymbol{E} + \hat{\boldsymbol{H}}\boldsymbol{G} & \boldsymbol{0} & \hat{\boldsymbol{G}}\boldsymbol{F} + \hat{\boldsymbol{H}}\boldsymbol{H} \end{pmatrix}, \tag{27}$$

which completes the proof. $\square$

**Lemma 3.** *Let $\boldsymbol{X}$ be a square matrix whose $i^{th}$ row and column are 0. The matrix exponential "ignores" these zeros. Specifically, suppose*

$$\boldsymbol{X} = \begin{pmatrix} \boldsymbol{E} & \boldsymbol{0} & \boldsymbol{F} \\ \boldsymbol{0} & 0 & \boldsymbol{0} \\ \boldsymbol{G} & \boldsymbol{0} & \boldsymbol{H} \end{pmatrix}, \qquad \exp_m\begin{pmatrix} \boldsymbol{E} & \boldsymbol{F} \\ \boldsymbol{G} & \boldsymbol{H} \end{pmatrix} = \begin{pmatrix} \bar{\boldsymbol{E}} & \bar{\boldsymbol{F}} \\ \bar{\boldsymbol{G}} & \bar{\boldsymbol{H}} \end{pmatrix}. \tag{28}$$

*Then we have*

$$\exp_m(\boldsymbol{X}) = \begin{pmatrix} \bar{\boldsymbol{E}} & \boldsymbol{0} & \bar{\boldsymbol{F}} \\ \boldsymbol{0} & 1 & \boldsymbol{0} \\ \bar{\boldsymbol{G}} & \boldsymbol{0} & \bar{\boldsymbol{H}} \end{pmatrix}. \tag{29}$$

*Proof.* The matrix exponential is given by

$$\exp_m(\boldsymbol{X}) = \sum_{k=0}^{\infty} \frac{1}{k!}\boldsymbol{X}^k. \tag{30}$$

By Lemma 2, all non-identity terms of this series are identical to terms in the series for $\exp_m\begin{pmatrix} \boldsymbol{E} & \boldsymbol{F} \\ \boldsymbol{G} & \boldsymbol{H} \end{pmatrix}$ but with zeros inserted in the $i^{th}$ row and column. The identity term then contributes the extra 1 in the main diagonal. $\square$

**Theorem 4.** *Let $\boldsymbol{U} \in \mathcal{M}$, and let $\tilde{\boldsymbol{U}}$ be its first three columns so that $\boldsymbol{U} = \begin{pmatrix} \tilde{\boldsymbol{U}} & \boldsymbol{a} \end{pmatrix}$.*

*The exponential on $\mathcal{M}$ can be computed as the exponential on $\mathrm{St}(n, 3)$ after discarding the last column of $\Delta = \begin{pmatrix} \tilde{\Delta} & \boldsymbol{0} \end{pmatrix}$.*

$$\exp_{\boldsymbol{U}}(\Delta) = \begin{pmatrix} \exp_{\tilde{\boldsymbol{U}}}(\tilde{\Delta}) & \boldsymbol{a} \end{pmatrix} \tag{31}$$

*Similarly, the logarithm on $\mathcal{M}$ can be computed as the logarithm on $\mathrm{St}(n, 3)$ followed by concatenating a zero column:*

$$\log_{\boldsymbol{U}_0}(\boldsymbol{U}_1) = \begin{pmatrix} \log_{\tilde{\boldsymbol{U}}_0}(\tilde{\boldsymbol{U}}_1) & \boldsymbol{0} \end{pmatrix} \tag{32}$$

*Proof.* We go line-by-line through Algorithm 1 to show that the Stiefel exponential on $\mathcal{M}$ is equivalent to the exponential for $\mathrm{St}(n, 3)$.

The last column of $(\boldsymbol{I}_n - \boldsymbol{U}\boldsymbol{U}^\top)\Delta$ is 0 because the last column of $\Delta$ is 0. Then,

$$\boldsymbol{Q}\boldsymbol{R} = (\boldsymbol{I}_n - \boldsymbol{U}\boldsymbol{U}^\top)\Delta = \begin{pmatrix} \tilde{\boldsymbol{Q}} & \boldsymbol{q} \end{pmatrix} \begin{pmatrix} \tilde{\boldsymbol{R}} & \boldsymbol{0} \\ \boldsymbol{0} & 0 \end{pmatrix}, \tag{33}$$

where $\boldsymbol{q}$ is some orthogonal vector to the 3 columns of $\tilde{\boldsymbol{Q}}$, and $\tilde{\boldsymbol{R}}$ is the first 3 columns and rows of $\boldsymbol{R}$. This corresponds to the QR decomposition $\tilde{\boldsymbol{Q}}\tilde{\boldsymbol{R}} = (\boldsymbol{I}_n - \tilde{\boldsymbol{U}}\tilde{\boldsymbol{U}}^\top)\tilde{\Delta}$.

In addition, we have that $\boldsymbol{U}^\top \Delta = \begin{pmatrix} \tilde{\boldsymbol{U}}^\top \tilde{\Delta} & \boldsymbol{0} \\ \boldsymbol{0} & 0 \end{pmatrix}$.

Thus, the block matrix $\boldsymbol{A}$ in Algorithm 1 is given as

$$\begin{pmatrix} \tilde{\boldsymbol{U}}^\top \tilde{\Delta} & \boldsymbol{0} & -\tilde{\boldsymbol{R}}^\top & \boldsymbol{0} \\ \boldsymbol{0} & 0 & \boldsymbol{0} & 0 \\ \tilde{\boldsymbol{R}} & \boldsymbol{0} & \boldsymbol{0} & \boldsymbol{0} \\ \boldsymbol{0} & 0 & \boldsymbol{0} & 0 \end{pmatrix}. \tag{34}$$

By Lemma 3, the matrix exponential of $\boldsymbol{A}$ "ignores" the extra 2 rows and 2 columns of zeros.

Therefore,

$$\left( \begin{array}{c} \boldsymbol{M}(t) \mid \cdots \\ \hline \boldsymbol{N}(t) \mid \cdots \end{array} \right) = \exp_m(t\boldsymbol{A}) = \exp_m t \begin{pmatrix} \tilde{\boldsymbol{U}}^\top \tilde{\Delta} & \boldsymbol{0} & -\tilde{\boldsymbol{R}}^\top & \boldsymbol{0} \\ \boldsymbol{0} & 0 & \boldsymbol{0} & 0 \\ \tilde{\boldsymbol{R}} & \boldsymbol{0} & \boldsymbol{0} & \boldsymbol{0} \\ \boldsymbol{0} & 0 & \boldsymbol{0} & 0 \end{pmatrix} = \left( \begin{array}{cc|c} \tilde{\boldsymbol{M}}(t) & \boldsymbol{0} & \cdots \\ \boldsymbol{0} & 1 & \cdots \\ \hline \tilde{\boldsymbol{N}}(t) & \boldsymbol{0} & \cdots \\ \boldsymbol{0} & 0 & \cdots \end{array} \right),$$

$$\tag{35}$$

where tilde terms are equal to their counterparts in the exponential of $\mathrm{St}(n, 3)$.

Finally, the output of Algorithm 1 is

$$\boldsymbol{U}\boldsymbol{M}(t) + \boldsymbol{Q}\boldsymbol{N}(t) = \begin{pmatrix} \tilde{\boldsymbol{U}} & \boldsymbol{a} \end{pmatrix} \begin{pmatrix} \tilde{\boldsymbol{M}}(t) & \boldsymbol{0} \\ \boldsymbol{0} & 1 \end{pmatrix} + \begin{pmatrix} \tilde{\boldsymbol{Q}} & \boldsymbol{q} \end{pmatrix} \begin{pmatrix} \tilde{\boldsymbol{N}}(t) & \boldsymbol{0} \\ \boldsymbol{0} & 0 \end{pmatrix} \tag{36}$$

$$= \begin{pmatrix} \tilde{\boldsymbol{U}}\tilde{\boldsymbol{M}}(t) & \boldsymbol{a} \end{pmatrix} + \begin{pmatrix} \tilde{\boldsymbol{Q}}\tilde{\boldsymbol{N}}(t) & \boldsymbol{0} \end{pmatrix} \tag{37}$$

$$= \begin{pmatrix} \tilde{\boldsymbol{U}}\tilde{\boldsymbol{M}}(t) + \tilde{\boldsymbol{Q}}\tilde{\boldsymbol{N}}(t) & \boldsymbol{a} \end{pmatrix} \tag{38}$$

$$= \begin{pmatrix} \exp_{\tilde{\boldsymbol{U}}}(\tilde{\Delta}) & \boldsymbol{a} \end{pmatrix}. \tag{39}$$

Since the Stiefel exponential is locally invertible, this also shows that the Stiefel logarithm can be computed using $\tilde{\boldsymbol{U}}_0$ and $\tilde{\boldsymbol{U}}_1$.

$\square$

**Theorem 5.** *Let $\boldsymbol{U} \in \mathcal{M}$, and let $\tilde{\boldsymbol{U}}$ be its first three columns so that $\boldsymbol{U} = [\tilde{\boldsymbol{U}}\ \boldsymbol{a}]$. Given $\boldsymbol{Z} \in \mathbb{R}^{n \times 3}$, the minimum-norm projection of $[\boldsymbol{Z}\ \boldsymbol{0}]$ onto $\mathrm{T}_{\boldsymbol{U}}\mathcal{M}$ is given by $[\pi(\boldsymbol{Z})\ \boldsymbol{0}]$, where*

$$\pi(\boldsymbol{Z}) = \tilde{\boldsymbol{U}}\,\mathrm{skew}(\tilde{\boldsymbol{U}}^\top \boldsymbol{Z}) + (\boldsymbol{I}_n - \boldsymbol{U}\boldsymbol{U}^\top)\boldsymbol{Z} \in \mathbb{R}^{n \times 3} \tag{40}$$

$$= \tilde{\boldsymbol{U}}\,\mathrm{skew}(\tilde{\boldsymbol{U}}^\top \boldsymbol{Z}) + (\boldsymbol{I}_n - \boldsymbol{U}\boldsymbol{U}^\top)\boldsymbol{Z} \in \mathbb{R}^{n \times 3} \tag{41}$$

*Note there is no tilde in the second term.*

*Proof.* We can rewrite

$$T_{\boldsymbol{U}}\mathcal{M} = \{[\tilde{\Delta} \, \mathbf{0}] \mid \tilde{\boldsymbol{U}}^\top \tilde{\Delta} + \tilde{\Delta}^\top \tilde{\boldsymbol{U}} = \mathbf{0}, \ \tilde{\Delta}^\top \boldsymbol{a} = \mathbf{0}\} \tag{42}$$

$$= \{[\tilde{\Delta} \, \mathbf{0}] \mid \tilde{\Delta} \in T_{\tilde{\boldsymbol{U}}}\mathrm{St}(n,3) \cap \mathrm{span}(\boldsymbol{a})^\perp\}, \tag{43}$$

Edelman et al. (1998) give an orthogonal projection of $\boldsymbol{Z}$ onto $T_{\tilde{\boldsymbol{U}}}\mathrm{St}(n,3)$ as:

$$\pi_1(\boldsymbol{Z}) = \tilde{\boldsymbol{U}}\,\mathrm{skew}(\tilde{\boldsymbol{U}}^\top \boldsymbol{Z}) + (\boldsymbol{I}_n - \tilde{\boldsymbol{U}}\tilde{\boldsymbol{U}}^\top)\boldsymbol{Z}, \tag{44}$$

where $\mathrm{skew}(\boldsymbol{A}) = \frac{1}{2}(\boldsymbol{A} - \boldsymbol{A}^\top)$. The orthogonal projection of $\boldsymbol{Z}$ onto $\mathrm{span}(\boldsymbol{a})^\perp$ is

$$\pi_2(\boldsymbol{Z}) = (\boldsymbol{I}_n - \boldsymbol{a}\boldsymbol{a}^\top)\boldsymbol{Z}. \tag{45}$$

We can check that $\pi_1$ and $\pi_2$ commute with $\pi = \pi_1 \circ \pi_2 = \pi_2 \circ \pi_1$. Hence, $\pi$ is an orthogonal projection onto $T_{\tilde{\boldsymbol{U}}}\mathrm{St}(n,3) \cap \mathrm{span}(\boldsymbol{a})^\perp$, as desired. $\square$

## C  EXPERIMENTAL DETAILS

### C.1  ARCHITECTURE

Explicitly, the neural network takes in moments $(P_X, P_Y, P_Z)$, time $t$, atom types $\boldsymbol{a}$, and coordinates $\boldsymbol{X}$, and outputs a Stiefel tangent vector. Moments are embedded using sinusoidal features with wavelength geometrically spaced from 0.0001 to 10,000. Time is similarly embedded but with a wavelength range from 0.001 to 1. We use a reflection-equivariant network. Note that given a reflection-*invariant* function $f$, the mapping $\boldsymbol{X} \mapsto \mathrm{sign}(\boldsymbol{X}) \odot f(\boldsymbol{X})$ is reflection-equivariant, where $\mathrm{sign}(\boldsymbol{X})$ gives the element-wise signs of $\boldsymbol{X}$, with $\mathrm{sign}(0) = 0$. Thus, the problem is reduced to constructing a network that is reflection-invariant with respect to the input coordinates $\boldsymbol{X}$.

As input to our network, we begin by featurizing the molecule in a reflection-invariant manner. We obtain invariant node features $\boldsymbol{h}_i \in \mathbb{R}^{d_{\mathrm{node}}}$ by using the molecule's unsigned coordinates and atom types, and edge features $\boldsymbol{e}_{ij} \in \mathbb{R}^{d_{\mathrm{edge}}}$ are computed from the unsigned differences between pairs of atomic coordinates. These features are passed through a Transformer backbone (Vaswani et al., 2017). We use the PreLN layout with an adaptive version of LayerNorm (Dieleman et al., 2022; Dhariwal & Nichol, 2021) that conditions on the timestep and molecule's moments. In addition, the attention module is replaced with a message-passing block that jointly updates the node and edge features:

$$(\boldsymbol{v}_{ij}, a_{ij}, \boldsymbol{e}'_{ij}) \leftarrow \mathrm{MLP}(\boldsymbol{h}_i, \boldsymbol{h}_j, \boldsymbol{e}_{ij}), \quad \text{for all } i, j, \tag{46}$$

$$\boldsymbol{e}_{ij} \leftarrow \boldsymbol{e}_{ij} + \boldsymbol{e}'_{ij}, \quad \text{for all } i, j, \tag{47}$$

$$\boldsymbol{y}_i \leftarrow \sum_{j=1}^{n} \left( \frac{\exp(a_{ij})}{\sum_{k=1}^{n} \exp(a_{ik})} \right) \boldsymbol{v}_{ij}, \quad \text{for all } i, \tag{48}$$

$$\boldsymbol{h}_i \leftarrow \boldsymbol{h}_i + \mathrm{Linear}(\boldsymbol{y}_i) \quad \text{for all } i. \tag{49}$$

The last two equations are reminiscent of self-attention in Transformers, and along the same lines, we use a multi-headed extension of them. Across all experiments, we use $d_{\mathrm{node}} = 768$ and $d_{\mathrm{edge}} = 192$. We use 16 Transformer-like blocks, 12 update heads, SiLU/Swish activations, and a $4\times$ expansion in each block's feed-forward module. In total, our model has 154M trainable parameters.

This architecture is distinct from the pretrained network of KREED (Cheng et al., 2024), which was tailored for predicting 3D structure given molecular formula, moments of inertia, and *substitution coordinates*. KREED was trained using random dropout of input substitution coordinates. For QM9, the model sometimes observed examples with no substitution coordinates during training. However, for GEOM, the model always received at least some substitution coordinates during training. Therefore, KREED is operating out-of-distribution when provided with no substitution coordinates at all on GEOM.

### C.2  STOCHASTICITY

Diversity is important for identifying unknown molecules outside the training set. Therefore, we experiment with adding stochasticity to the dynamics of the flow, leading individual paths to be stochastic. To do so during training, we apply an exponential map to a Gaussian variable with noise scale $\gamma \cdot \frac{\cos(\pi t)+1}{2}$ in the tangent space of the interpolant before calculating $\dot{\boldsymbol{U}}$ again. During sampling, we add Gaussian noise of the same noise schedule to the tangent vector at every time step.

Table 3: General training and sampling hyperparameters.

| | Hyperparameter | QM9 | GEOM |
|---|---|---|---|
| Training | Epochs | 1000 | 60 |
| | Batch size per GPU | 256 | 24 |
| | Optimizer | AdamW | AdamW |
| | Learning rate | $10^{-4}$ | $10^{-4}$ |
| | Learning rate warmup steps | 2000 | 2000 |
| | Weight decay | 0.01 | 0.01 |
| | Gradient clipping | yes | yes |
| | EMA decay | 0.9995 | 0.9995 |
| KREED | Timesteps | 1000 | 1000 |
| | Schedule | polynomial | polynomial |
| Stiefel FM | Timesteps | 200 | 200 |

Table 4: Training and sampling hyperparameters for Stiefel Flow Matching.

| Dataset | Model | Timestep sampling | OT | stochasticity $\gamma$ |
|---|---|---|---|---|
| QM9 | Stiefel FM | uniform | no | 0.00 |
| | Stiefel FM-OT | uniform | yes | 0.00 |
| | Stiefel FM-OT-stoch | uniform | yes | 0.10 |
| | Stiefel FM-ln | logit-normal | no | 0.00 |
| | Stiefel FM-ln-OT | logit-normal | yes | 0.00 |
| GEOM | Stiefel FM | uniform | no | 0.00 |
| | Stiefel FM-OT | uniform | yes | 0.00 |

Table 5: Extended ablation study for QM9. Stochasticity and logit-normal timestep sampling do not provide a clear advantage.

| Method | % < RMSD ↑ | | Error ↓ | Valid ↑ | Stable ↓ | Diverse ↑ | NFE ↓ |
|---|---|---|---|---|---|---|---|
| | 0.25 Å | 0.10 Å | | | | | |
| Stiefel FM | **15.17 ± 0.31** | **13.82 ± 0.30** | 0.00 | 0.882 | −1.125 | 1.040 | 200 |
| Stiefel FM-OT | 13.99 ± 0.30 | 12.68 ± 0.29 | 0.00 | 0.835 | −1.039 | 1.045 | 200 |
| Stiefel FM-stoch | **15.13 ± 0.31** | **13.83 ± 0.30** | 0.00 | 0.877 | −1.116 | 1.045 | 500 |
| Stiefel FM-ln | **15.74 ± 0.32** | 11.45 ± 0.28 | 0.00 | 0.880 | −0.600 | 0.982 | 200 |
| Stiefel FM-ln-OT | 14.90 ± 0.31 | 12.45 ± 0.29 | 0.00 | 0.875 | −0.687 | 1.026 | 200 |

| Method | Dataset | Training (min / epoch) | Training (it / s) | Sampling (seconds / K=10 samples) |
|---|---|---|---|---|
| KREED-XL | QM9 | 1.02 | 6.7 | 13.9 |
| Stiefel FM | QM9 | 1.32 | 5.1 | 2.9 |
| Stiefel FM-OT | QM9 | 3.48 | 1.9 | 2.9 |
| KREED-XL | GEOM | 225.6 | 17.0 | 71.3 |
| Stiefel FM | GEOM | 224.4 | 17.1 | 15.0 |
| Stiefel FM-OT | GEOM | 229.8 | 16.7 | 15.0 |

Table 6: Training and sampling timing on QM9 and GEOM. A CPU bottleneck of computing logarithms and optimal transport exists for QM9 due to a large batch size of 256, but this CPU bottleneck disappears for GEOM due to a small batch size of 24.

## C.3 TRAINING

Models were trained on 4 NVIDIA A100 40GB GPUs. Tables 3 and 4 gives our hyperparameters. We use the adaptive gradient clipping strategy from Hoogeboom et al. (2022).

## C.4  SAMPLING

During sampling, we sample uniformly from $\mathcal{M}$ and iteratively query the trained model for a tangent vector $\Delta$ at every step. If stochasticity is turned on, Gaussian noise with scale $\gamma \cdot \frac{\cos(\pi t)+1}{2}$ is added to this tangent vector. At every step, $\boldsymbol{U}_t$ is projected onto the manifold (Appendix B.6) before projecting the tangent vector to the tangent space on the manifold $\mathrm{T}_{\boldsymbol{U}_t}\mathrm{St}(n,4)$ (Theorem 5). Integration proceeds by applying $\exp_{\boldsymbol{U}_t}$ to this tangent vector, scaled by $dt$.

---

**Algorithm 3** Sampling under Stiefel Flow Matching.

---

**Require:** Size $n$, atom types $\boldsymbol{a}$, moments $P_{XYZ} = (P_X, P_Y, P_Z)$, stochasticity $\gamma$, timesteps $T$
1: $\boldsymbol{U} \sim \mathrm{Uniform}(\mathrm{St}(n,4))$ using Appendix B.4
2: Project $\boldsymbol{U}$ to $\mathcal{M}$ (defined in Equation 5) using Appendix B.5
3: $t \leftarrow 0$
4: $t_\Delta \leftarrow 1/T$
5: **for** step in $1, \ldots, T$ **do**
6: $\quad$ $\boldsymbol{X} \leftarrow \boldsymbol{X}(\boldsymbol{U}, \boldsymbol{m}, P_{XYZ})$ by inverting Equation 4
7: $\quad$ $\Delta \leftarrow v_\theta(t, \boldsymbol{X}, P_{XYZ}, \boldsymbol{a}) \in \mathbb{R}^{n \times 3}$
8: $\quad$ **if** step $< T$ **then**
9: $\quad\quad$ $\Delta \leftarrow t_\Delta \Delta + \frac{1}{2}\gamma\sqrt{t_\Delta}(\cos(\pi t)+1)\boldsymbol{\varepsilon}$, where $\varepsilon_{ij} \sim \mathcal{N}(0,1)$
10: $\quad$ **else**
11: $\quad\quad$ $\Delta \leftarrow t_\Delta \Delta$
12: $\quad$ Project $\Delta$ to $\mathrm{T}_{\boldsymbol{U}}\mathcal{M}$ using Theorem 5
13: $\quad$ $\boldsymbol{U} \leftarrow \exp_{\boldsymbol{U}}(\Delta)$
14: $\quad$ $t \leftarrow t + t_\Delta$
$\quad$ **return** $\boldsymbol{X} \leftarrow \boldsymbol{X}(\boldsymbol{U}, \boldsymbol{m}, P_{XYZ})$ by inverting Equation 4

---

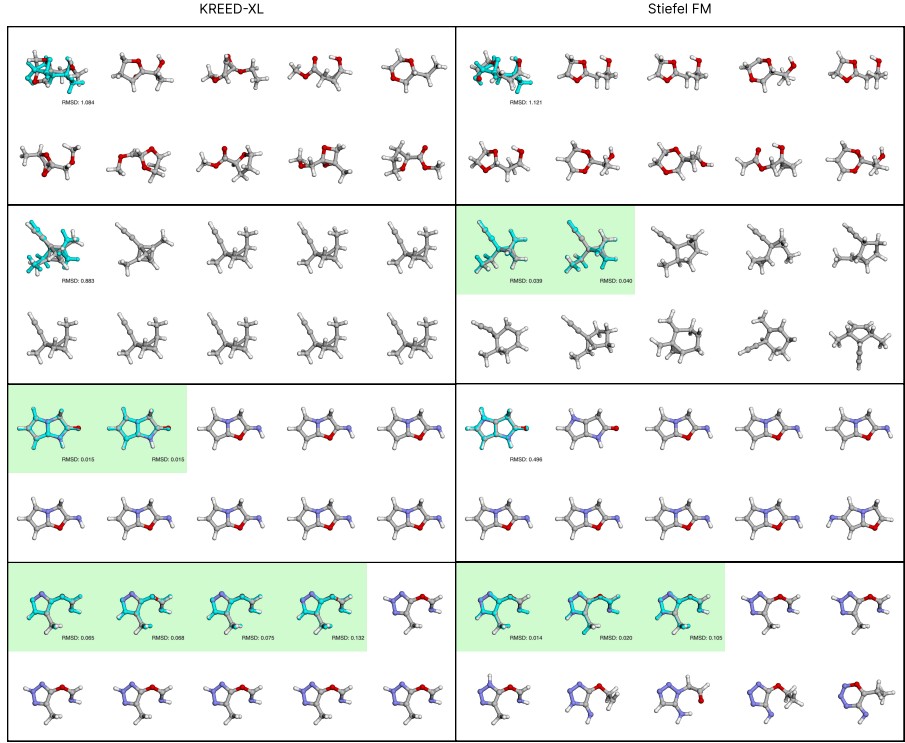

Figure 5: Selected QM9 examples. Best viewed zoomed in. Examples are sorted by RMSD to ground truth, which is shown in cyan. Green panels indicate meeting the threshold of 0.25 Å.

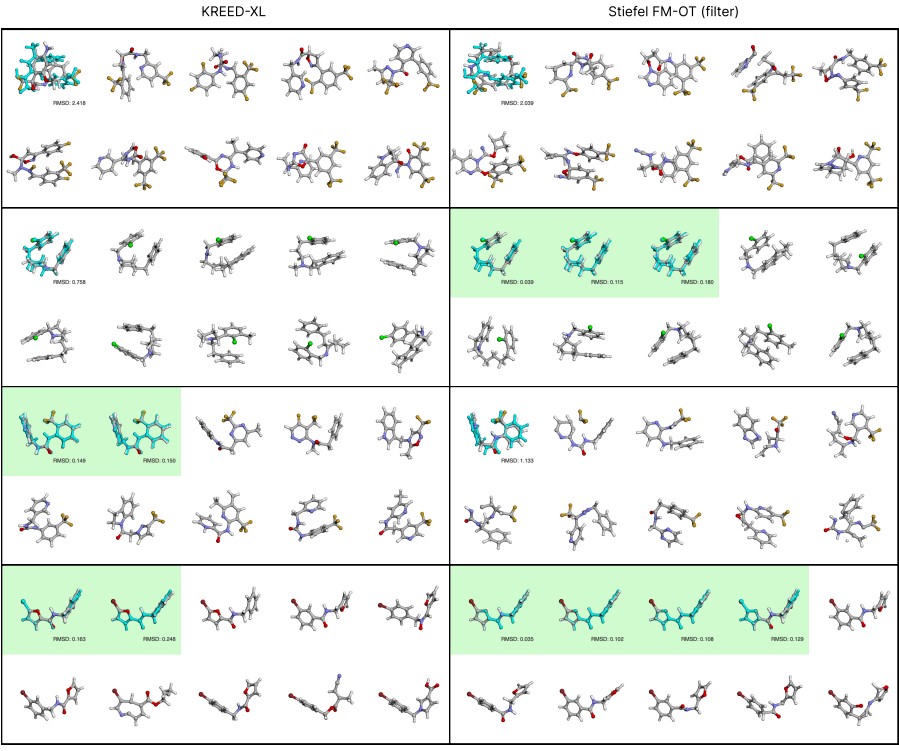

Figure 6: Selected GEOM examples. Best viewed zoomed in. Examples are sorted by RMSD to ground truth, which is shown in cyan. Green panels indicate meeting the threshold of 0.25 Å.

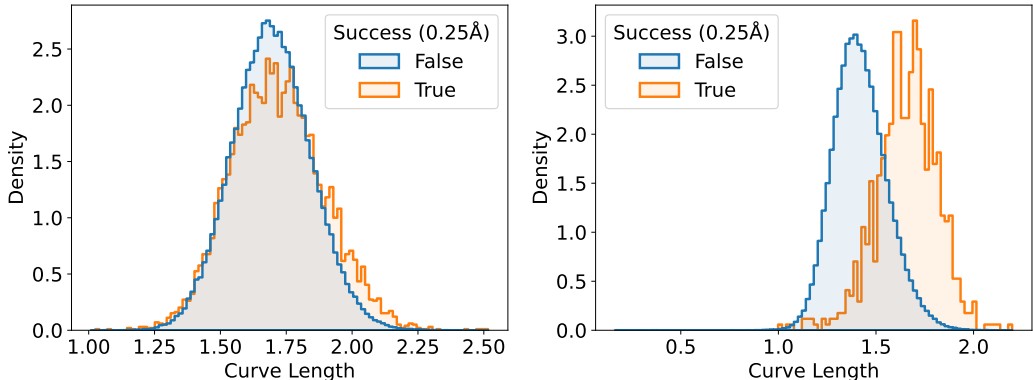

Figure 7: Normalized histogram of curve lengths of success and failure examples of Stiefel FM on the QM9 (left) and GEOM (right) datasets. Success cases are more often the result of longer generation paths.

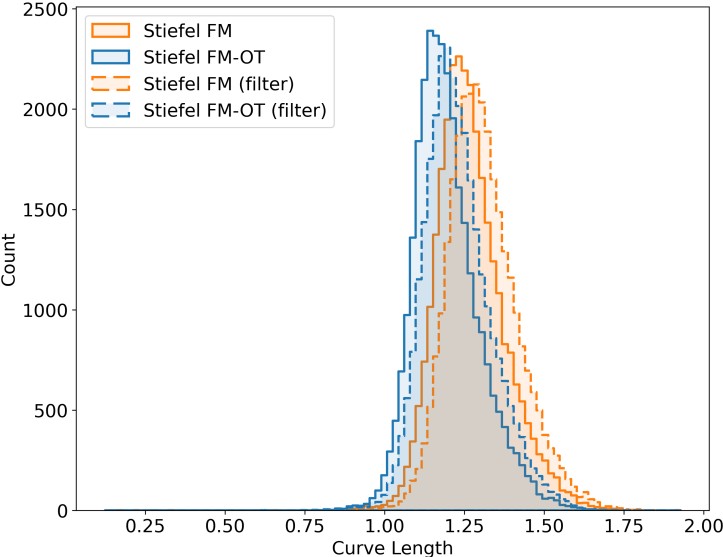

Figure 8: Equivariant optimal transport shortens generation trajectories for GEOM. However, generating more samples and filtering by validity has a slight bias towards *longer* generation trajectories.

## D  GREEDY RANDOM OPTIMAL ASSIGNMENT ALGORITHM

The greedy random local search looks for the best reflection and permutation to minimize the Stiefel distance between a given $U_0$ and $U_1$, $d(U_0, U_1) = ||\log_{\Pi U_0 R}(U_1)||_{\Pi U_0 R}$. This distance is approximated using only one iteration of the logarithm. It first decides on the best reflection $R$ by trying many random permutations for each reflection. Then, the best reflection is kept, and the permutation is optimized using random swaps of indices. The computational cost of computing the optimal transport map is $O(np^2)$, as it relies on a fixed number of computations of the logarithm.

---

**Algorithm 4** Heuristic alignment algorithm.

---

**Require:** $U_0, U_1 \in \mathrm{St}(n, p)$, atom types $a$, number of restarts $R$, local search budget $L$
  1: $c^* \leftarrow \infty$ ⊳ lowest cost seen so far
  2: **for** reflections $R \in \{\mathrm{diag}(s) \mid s \in \{-1, +1\}^3\}$ **do** ⊳ find a good reflection
  3:     **for** $k = 0, 1, \ldots, R$ **do**
  4:         sample a random atom-type-preserving node permutation $\Pi$
  5:         $U_{\mathrm{cand}} \leftarrow \Pi U_0 R^\top$
  6:         $c \leftarrow \tilde{d}(U_{\mathrm{cand}}, U_1)$, ⊳ approximate distance
  7:         **if** $c < c^*$ **then**
  8:             $c^* \leftarrow c$
  9:             $U_{\mathrm{best}} \leftarrow U_{\mathrm{cand}}$
 10: **for** $k = 0, 1, \ldots, L$ **do** ⊳ local search over node swaps
 11:     sample $(i, j)$ such that $a_i = a_j$, without repeating pairs
 12:     $U_{\mathrm{cand}} \leftarrow \mathrm{swap}(U_{\mathrm{best}}, i, j)$
 13:     $c \leftarrow \tilde{d}(U_{\mathrm{cand}}, U_1)$, ⊳ approximate distance
 14:     **if** $c < c^*$ **then**
 15:         $c^* \leftarrow c$
 16:         $U_{\mathrm{best}} \leftarrow U_{\mathrm{cand}}$
 17: **return** $U_{\mathrm{best}}$

---

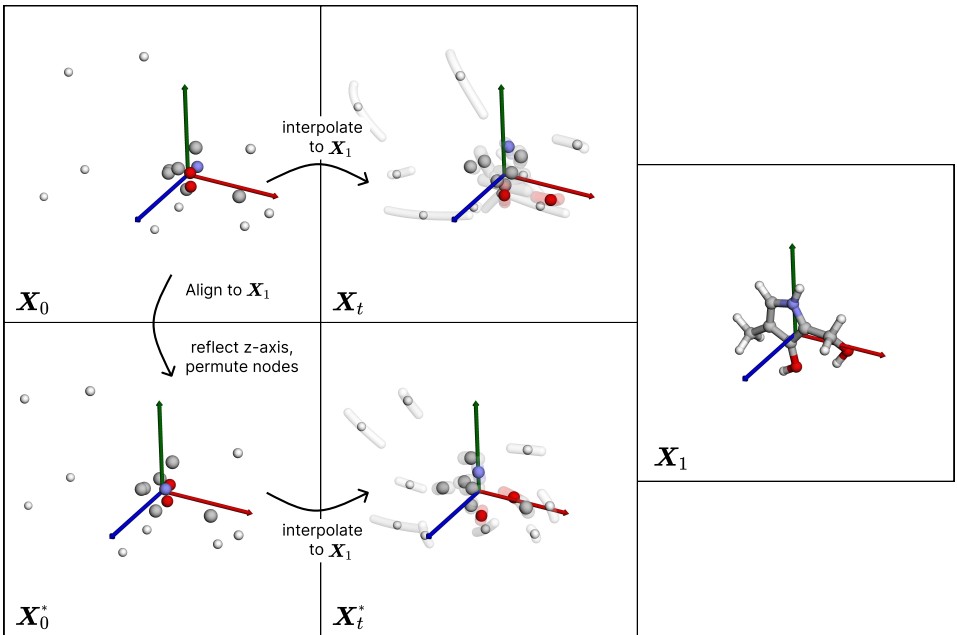

Figure 9: Equivariant optimal transport aligns noise samples $X_0$ to training examples $X_1$ over reflections and permutations, leading to smoother and shorter paths shown to the model during training.

## E   STIEFEL LOGARITHM EMPIRICAL ANALYSIS

We empirically analyze the convergence of the Stiefel logarithm (Algorithm 2). We sample 100k training examples from QM9 or GEOM to be used as $U_1$ and for each example sample one random point $U_0$. We compute the true logarithm $\Delta = \log_{U_0}(U_1)$ using a large number of iterations and compare it to the 20-iteration truncated logarithm. We record error as the infinity norm of the difference between the true and approximate logarithms. We set the convergence threshold to be 1e-6.

For QM9, the 20-iteration logarithm converges 97.8% of the time (median 9 iterations to converge), and the median error in case of nonconvergence is 2.5e-4. The Spearman correlation between the 1-iteration approximate distance and the true distance is $\rho = 0.87$. For GEOM, the 20-iteration logarithm converges 99.7% of the time (median 9 iterations to converge), and the median error in case of nonconvergence is 8.8e-5. The Spearman correlation between the 1-iteration approximate distance and the true distance is $\rho = 0.83$.

These results empirically validate that the 1-iteration approximate distance used in Algorithm 4 is an upper bound on the true distance, and that it is a valid heuristic which generally maintains the same relative ordering as the true Stiefel distance.

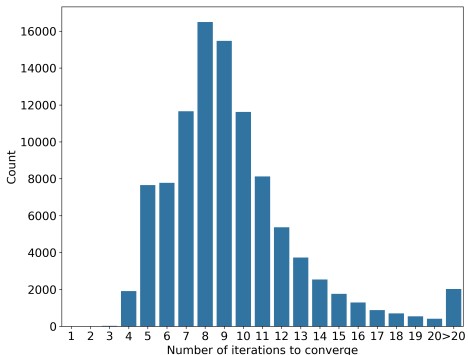 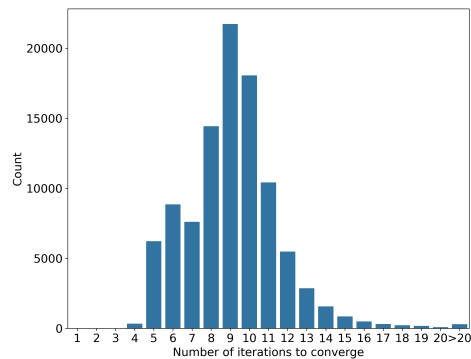

Figure 10: Histograms of the number of inner iterations of the Stiefel logarithm required to converge to an error of 1e-6 for 100k random training examples of QM9 (left) and GEOM (right).

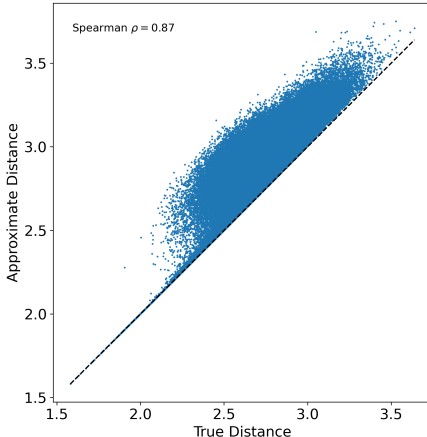 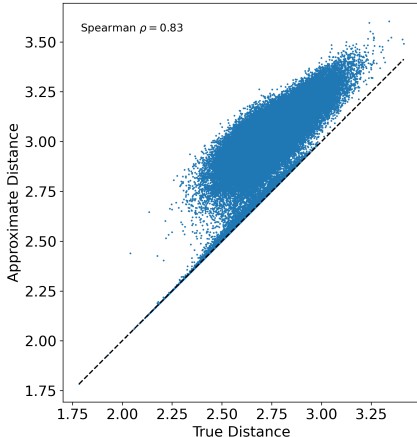

Figure 11: Parity plots comparing the 1-iteration approximate Stiefel distance to the true Stiefel distance of QM9 (left) and GEOM (right).

