# OpenReview forum: "Stiefel Flow Matching for Moment-Constrained Structure Elucidation"
_ICLR.cc/2025/Conference — ICLR 2025 Poster_

### Official Review · Reviewer_519h · 2024-10-21

**Soundness:** 3
**Presentation:** 4
**Contribution:** 3
**Rating:** 8
**Confidence:** 4

**Summary:**

This work proposed a novel flow-based generative framework to address the generation task of molecules with moment constraints. The authors noted the connection between the molecules with fixed moments and the Stiefel manifold and applied flow matching on such a manifold to ensure exact moment constraints. Empirical experiments demonstrated better generation results that satisfy these moment constraints.

**Strengths:**

- The manifold structure of n-body particles with fixed moments was demonstrated to be a Stiefel manifold in the paper, which, to the best of my knowledge, is the very first work.
- Equipped with such a Riemannian structure, the proposed framework can effectively navigate through the Stiefel manifold with the moment constraints exactly satisfied. Empirical experiments also demonstrated zero errors against the moment constraints.
- A thorough mathematical introduction to the Stiefel manifold was provided in the Appendix, together with efficient algorithms to calculate the exponential and logarithm maps on the Stiefel manifold, which makes practical training feasible.

**Weaknesses:**

1. The idea of the moment-constrained generation is not well-motivated in the paper.
    - **Molecules are not rigid bodies**. The dynamics of a molecule (e.g., stretching, twisting of bonds) make it possible for it to have various conformations that vary slightly by their principal moment of inertia. In other words, the moment of inertia measured by rotational spectroscopy is an ensemble property that reflects the average (or effective) moment of inertia of a molecule. In this sense, it seems more reasonable to allow for some violation of the moment constraints.
    - **Practical applications are missing**. It remains unclear what the practical applications of generating molecules with specific moments are. As an example, generating molecules with a specific HOMO-LUMO gap can be significant to the discovery of photoelectric molecules. The moment of inertia, however, does not hold a specific connection to real-world applications.

2. **The number of baselines compared in the paper was small**. Indeed, only KREED was compared. There are other baselines available:
    - The conditional generative model in [1] where molecular properties are fed as additional information during both training and sampling for the conditional generation of molecules with desired properties. As the generative model implicitly learns the connection between the property and the molecule, it does not guarantee the exact satisfaction of constraints, but in practice, it does provide fairly competitive generation results.
    - As it is always possible to project the generation to the corresponding Stiefel manifold according to Appendix B.6, another natural baseline would be directly projecting the unconstrained generation onto the Stiefel manifold, which also ensures the exact moment constraints.

3. **The experimental results were not convincing enough** to demonstrate the superior performance of the proposed method. Although the moment constraint errors were indeed zero, the validity and stability scores were worse. The RMSD percentage also outperformed the baseline by only a small margin on the GEOM dataset.

4. **The filtering procedure on the GEOM evaluation may give the proposed model with inappropriate advantages**, as it is essentially picking from a larger number of generations with a high likelihood of finding better samples. A similar procedure should be applied to the KREED baseline for a fair comparison.

[1] Hoogeboom, Emiel, et al. "Equivariant diffusion for molecule generation in 3d." International conference on machine learning. PMLR, 2022.

**Questions:**

1. What are the practical applications of the moment-constrained molecule generation? Why is it important to enforce the exact moment constraints? See Weakness 1.
2. What are the performance of additional baselines? See Weakness 2.
3. Can you follow the same postprocessing procedure and compare the proposed model with the KREED-filter baseline performance? See Weakness 3 & 4.
4. What is the empirical training and sampling time of the proposed method compared to a normal unconstrained Euclidean flow matching model? I would expect extra time as the exponential map and logarithm map on the Stiefel manifold are far more complex to compute. Nonetheless, it is always beneficial to know the empirical time.
5. The logarithm map was calculated with 20 iterations. Does it provide a good enough approximation? Can you provide an analysis of the logarithm errors with respect to the number of iterations to better demonstrate your choice of 20 iterations (and 1 iteration for OT)?

---

> ### Author Response · Authors · 2024-11-22
>
> Thank you for your thoughtful comments. We appreciate your recognition of the novelty of our work in using the Stiefel manifold to exactly satisfy moment constraints and in providing a practical training procedure for an effective Stiefel generative model. We address your comments point-by-point below.
>
> **Molecules are not rigid bodies.** You are correct that molecules are not perfectly rigid, and that naively converting rotational constants to moments yields *effective moments*, not equilibrium moments. We have added a discussion of vibrational flexibility and rovibrational correction in Appendix A.4. In summary, exact moment constraints enables the search of the true equilibrium moments *up to the precision of calculating the rovibrational correction*.
>
> Experiment observes properties that have been *vibrationally* averaged, including the rotational constants $A(BC)_0$. Vibrational averaging is distinguished from conformational fluctuations such as torsions, which can be frozen out by cooling molecules to their ground vibrational state. However, even in the ground vibrational state, a molecule is still vibrating due to zero-point energy.
> If one were to *a priori* guess the equilibrium moments correctly, one could then verify whether they are indeed correct by generating structures, calculating their rovibrational corrections $\alpha$, and then *checking* their agreement to the experimental rotational constants, via $A(BC)_0 = A(BC)_e - \frac{1}{2}\sum_i \alpha_i^{A(BC)}$.
> Therefore, given experimental rotational constants $A(BC)_0$, one can use this verification procedure in a fine-grid search for the true equilibrium rotational constants $A(BC)_e$. This is feasible since $A(BC)_e$ consist of only 3 numbers. Experimental precision in $A(BC)_0$ is maintained up to the precision in computing $\alpha$.
>
> To verify that a structure is the true structure, we must know that:
> 1. its $A(BC)_e$ and $\alpha$ match $A(BC)_0$
> 2. gradient norm is near 0.
>
> In contrast, without the manifold constraint, samples may have incorrect moment constraints, and we must rely on quantum chemistry geometry optimization to land in a potential energy minimum that just happens to have the correct rovibrational-corrected moments.
>
> **Practical application.** The practical application is in enabling unknown molecule structure elucidation by rotational spectroscopy, as rotational spectroscopy can measure rotational constants, which are related to the equilibrium moments once you know the rovibrational correction. Our reported success rates are necessarily low due to the need to evaluate on many different test set molecules - in practice, a structure elucidation campaign focuses on a handful of molecules and can therefore devote large computational resources (e.g. 100x more samples, exhaustive energy calculations).
>
> An immediate practical implication of our results is that it actually is possible to identify unknown molecules by just their moments of inertia and molecular formula. The experimental implication of solving this problem is that it could dramatically shift the role of rotational spectroscopy from merely confirming structures to elucidating unknown molecules. This would provide a new method for structure elucidation which does not require mixture separation and may be useful in identifying unknown molecules in the interstellar medium.
>
> **Number of baselines is small.** See global response. Thank you for the suggestion of KREED-XL-proj.
>
> **Experimental results not convincing.** We emphasize that validity and stability are reported as averages over *all generated samples*. Despite poorer averages, Stiefel FM-OT (filter) obtains a greater success rate. The fact that an underfitting model can already provide advantage suggests that the direction of the approach is meaningful, and we discuss potential improvements that overcome the pathologies of Riemannian flow matching in the response to Reviewer *Locv*.

---

> > ### Author Response · Authors · 2024-11-22
> >
> > **Filtering provides advantage.** We argue that validity-filtering does not provide an unfair advantage because the filtering procedure does not provide access to the true RMSD oracle - it just filters out structures with steric clashes. It is not guaranteed that validity-filtering will improve performance because a model could instead generate more valid structures that are not the true structure. The fact that Stiefel FM-OT can effectively leverage the validity filter demonstrates that exact moment constraints can help sift out the true structure.
> >
> > Put another way, the improved performance suggests that Stiefel FM (filter), when adjusted for generating valid structures, is more likely to select out the valid structure that actually is the true structure. This is in contrast to KREED-XL, which can readily generate valid structures, but which may not be the correct structure.
> >
> > Lastly, Stiefel FM-OT (filter) still has a lower computational cost than KREED-XL: we report empirical timing of sampling below. Though KREED-XL could be adapted to Euclidean flow matching to reduce NFE cost, it is not clear that Euclidean flow matching would maintain its current success rate, since stochastic sampling empirically leads to higher quality samples [1], and this structure elucidation task particularly calls for low-temperature sampling. It is not clear how to fairly perform a filtering procedure for KREED-XL due to its much larger computational cost: KREED-XL requires 71.3 seconds per example, whereas Stiefel FM-OT (filter) requires 45.0 seconds per example.
> >
> > [1] Karras, T., Aittala, M., Aila, T., & Laine, S. (2022). Elucidating the design space of diffusion-based generative models. Advances in neural information processing systems, 35, 26565-26577.
> >
> > **Training and sampling overhead of Stiefel FM.** We summarize wall-clock timing on QM9 and GEOM below:
> >
> > The cost of Euclidean flow matching can be estimated by dividing the sampling cost of KREED-XL by 5, though success rate may not remain the same, for the reasons mentioned above.
> >
> > | Method | Dataset | Training (min / epoch) | Training (it / s) | Sampling (seconds / K=10 samples) |
> > | --- | --- | --- | --- | --- |
> > |KREED-XL | QM9 | 1.02 | 6.7 | 13.9 |
> > |Stiefel FM | QM9 | 1.32 | 5.1 | 2.9 |
> > |Stiefel FM-OT | QM9 | 3.48 | 1.9 | 2.9 |
> > |KREED-XL | GEOM | 225.6 | 17.0 | 71.3 |
> > |Stiefel FM | GEOM | 224.4 | 17.1 | 15.0 |
> > |Stiefel FM-OT | GEOM | 229.8 | 16.7 | 15.0 |
> >
> > The CPU bottleneck of logarithm computation and optimal transport disappears for GEOM due to a smaller batch size of 24, compared to a batch size of 256 for QM9.
> >
> > **Logarithm convergence.** Most examples converge before reaching 20 iterations. Only 1 iteration is used for OT because it is used as a heuristic for search, and it maintains the same relative ordering. We empirically validate this below.
> >
> > We sample 100k training examples from QM9 or GEOM to be used as $\boldsymbol{U}_1$ and for each example sample one random point $\boldsymbol{U}_0$. We compute the true logarithm using a large number of iterations and compare it to the 20-iteration truncated logarithm. We record error as the infinity norm of the difference between the true and approximate logarithms. We set the convergence threshold to be 1e-6.
> > - QM9: The 20-iteration logarithm converges 97.9% of the time (median 9 iterations to converge), and the median error in case of nonconvergence is 2.4e-4. The Spearman correlation between the 1-iteration approximate distance and the true distance is $\rho=0.88$.
> > - GEOM: The 20-iteration logarithm converges 99.7% of the time (median 9 iterations to converge), and the median error in case of nonconvergence is 8.8e-5. The Spearman correlation between the 1-iteration approximate distance and the true distance is $\rho=0.82$.
> >
> > These results also empirically show that the 1-iteration approximate distance is an upper bound on the true distance.
> >
> > Plots and histograms are shown in Appendix E.

---

> > > ### Comment · Reviewer_519h · 2024-11-25
> > > **Decision to raise my scores**
> > >
> > > I thank the authors for their detailed explanation regarding my concerns on the practical application of moment-constrained generation in the chemistry domain. With the additional baseline results and further ablation studies on the iterative algorithm for the exponential and logarithm maps, I believe the authors have fully addressed my concerns. Therefore, I decide to raise my score from 5 to 8 and champion this paper for its rigorous mathematical background and good experimental results.

---

### Official Review · Reviewer_oa92 · 2024-10-28

**Soundness:** 3
**Presentation:** 3
**Contribution:** 3
**Rating:** 6
**Confidence:** 2

**Summary:**

This paper proposes the use of flow matching generative models to obtain the 3D structure of molecules. This is achieved by explicitly enforcing problem constraints using Stiefel manifold and its properties.

**Strengths:**

- The paper is generally well-written and easy to follow. The math is sound and clearly presented.

- The use of Stiefel manifold and its connection to the problem and the constraints in (2).

- The use of geodesics in the Stiefel manifold that provides straight lines for navigating the manifold.

**Weaknesses:**

- How generalizable is the proposed method? Can the authors perform OOD evaluation?

- Why the need to generate K=10 samples and report only the one with the lowest RMSD? Is this the standard practice? if yes, support is needed. If not, then performing this indicates instability which requires further investigation.

- The evaluation metric for the proposed method seems to depends on RMSD thresholds defined by the authors. Have these thresholds been employed by previous methods that utilize neural networks for predicting the 3D structure of molecules? If yes, then citations are needed. The authors in Cheng et al. (2024) used correctness and reported the exact RMSD values.

- Why not reporting the RMSD directly? For example, when predicting a protein 3D structure (e.g., in AlphaFold), the RMSD is reported directly after alignment.

**Questions:**

### Major Comments/Questions:

- In lines 175 to 177, how practical is the assumption $n\geq 5$? How many examples with $n<5$ are removed from the datasets?

- To compute the loss in (8), the authors listed 4 requirements. All are well explained other than the second. Further clarification is needed here.

- Many details of the paper is put in the Appendix. This is disrupting the flow of the paper. I suggest that the authors to place an algorithmic procedure for the sampling part.

### Minor Comments/Questions:

- Based on the definition of $\mathbf{m}$, shouldn't the total number of masses $M$ be equal to $n$?

- Vector $\mathbf{a}$ is not being used in the problem definition other than specifying the number of atoms.

- The use of $st(n,3)$ instead of $st(n,4)$ in the main body is not well justified. Why Theorem 4 is put in the Appendix?

- The sampling procedure is not as clearly presented as the training of $v_\theta$. Methmatical procedure of how the pre-trained $v_\theta$ is employed to perform sampling is needed.

- In practice, the authors use t=0.001 to 1, but the uniform distribution in (8) uses U[0,1]. Shouldn't this be U(0,1]?

- The last sentence in the Datasets paragraph in Section 4 requires support by a citation.

- Why the FM model has more parameters than the diffusion model in Cheng et al. (2024)?

- Ill-sentence in lines 445 to 446.

---

> ### Author Response · Authors · 2024-11-22
>
> Thank you for your helpful comments. We appreciate you saying that the paper is well written and easy to follow, as well as your recognition of the novelty of our method in applying the Stiefel manifold and its geodesics. We address your comments below.
>
> **Generalizability of the method.** Our results are on a *molecule-wise* train/test split, which means that at test time, the model has never seen any of the unknown molecule's conformers. The increased success rate of Stiefel FM indicates that the model generalizes better than unconstrained diffusion models. We note that the GEOM dataset covers a large area of chemical space which also covers the extent of applicability of rotational spectroscopy: medium-size molecules. Out-of-distribution detection is not perceived to be an issue, since any reported successes will be checked thoroughly using quantum chemistry. Does this answer your question?
>
> **Generating $K=10$ samples.** We take the minimum RMSD because the important metric is whether the generative model can generate the correct structure at least once. We set $K=10$ based on our computational budget of requiring evaluation on all 29203 test set examples. In actual structure elucidation campaigns, there will only be a handful of molecules of interest, so $K$ should be $>1000$. Correspondingly, significant computational resources will be available so that every generated example can be checked by geometry optimization with quantum chemistry, followed by ensuring that the molecule is stable (gradient norm 0) and has the correct moments.
>
> We emphasize that we are proposing a new problem setting. Other 3D structure prediction problems such as docking usually query $K$ samples and then rank samples based on a confidence head to output a single top-1 predicted structure. We do not rank samples because every generated sample can be evaluated with quantum chemistry.
>
> **RMSD thresholds.** In our proposed problem setting, the RMSD thresholds are set quite small because the only way to verify that a generated sample is correct is to check that it is stable and has the correct moments. Stability is highly sensitive to tiny changes in conformation. The correctness metric of Cheng et al. is too forgiving, because torsion angles can be rotated, giving the same chemical graph but a different conformer with different moments of inertia.
>
> **Report RMSD directly.** We do not report mean/median RMSD because these aggregate metrics do not distinguish the performance of each method. Most examples have high RMSD, owing to the high difficulty of the problem, which washes out the mean or median. We therefore show RMSD histograms for transparency on the choice of threshold.
>
> **$n \geq 5$ assumption.** There are 6 examples of molecules with $n < 5$ in QM9, and none in GEOM. There are relatively very few small molecules compared to larger molecules. The vast majority of already-known molecules are also small, because they are easy to study. In contrast, we focus on the setting of elucidating unknown molecules.
>
> **Explaining loss computation.** Step (2) is geodesic interpolation, which is given by Eq. (9). We refer to the Appendix for numerical algorithms of the exponential and logarithm.
>
> **Appendix interrupts flow, add sampling.** We have done our best to make the main text of the paper self-contained in conveying important details related to method and experimental results. We intentionally leverage the appendix for showing greater details for interested readers. We are happy to discuss which specific details you think would be a better fit for the main text. We have also added an algorithm box describing sampling as you suggested.
>
> **Masses.** You are correct that there are $n$ masses, one for each atom. $M=\sum_i \boldsymbol{m}_i$ is the total mass of the molecule.
>
> **Vector $\boldsymbol{a}$.** This is defined this way just so it is clear that atom types are discrete. There is an implicit mapping between atom types and the mass of the most abundant isotope. $\boldsymbol{a}$ is used in the appendix to specify that input features to the neural network include the embedded discrete atom types.

---

> > ### Author Response · Authors · 2024-11-22
> >
> > **$\mathrm{St}(n, 3)$ vs $\mathrm{St}(n, 4)$.** The true degrees of freedom lie in $\mathrm{St}(n-1, 3)$, with a dimension of $3n-9$, accounting for $3n$ coordinates and removing 3 translational, 3 rotational, and 3 moment degrees of freedom. $\mathrm{St}(n-1, 3)$ is contained within $\mathrm{St}(n, 3)$, which in turn is contained within $\mathrm{St}(n, 4)$. The reason we consistently use $\mathrm{St}(n, 4)$ in the main text is to be specific about "which" $\mathrm{St}(n, 3)$ we are referring to, as it must maintain orthogonality to the mass vector. We only refer to $\mathrm{St}(n, 3)$ two times in the main text: (1) when we refer to the first 3 columns of $\boldsymbol{U}$, which does lie in $\mathrm{St}(n, 3)$; and (2) to say that the logarithm of $\mathcal{M} \subseteq \mathrm{St}(n, 4)$ can be computed using the logarithm of $\mathrm{St}(n, 3)$. Theorem 4, which proves this, is postponed to the Appendix because its consequences are just that the logarithm can be computed slightly faster.
> >
> > **Sampling procedure is not clear.** We include the ODE to be solved by sampling in Section 3 of the main text, and we include an algorithm box in the appendix.
> >
> > **Timestep embedding.** We clarify that the appendix refers to *embedding* the timestep $t$ for the neural network using sinusoidal features with a *wavelength range* from 0.001 to 1. Anyways, we empirically do sample $U(0,1)$ without the endpoints to avoid potential numerical issues, so we update this in the text.
> >
> > **Require citation: Low-energy conformer has high population.** In rotational spectrometers, the molecular sample is usually cooled before measurement, which makes it a reasonable assumption that the lowest-energy conformer has the highest population. We have added a citation for this.
> >
> > **More parameters.** Our model is trained with more parameters because of the expected increased difficulty of learning a velocity on the Stiefel manifold, rather than in Euclidean space. We ensure fairness by training a reflection-equivariant diffusion model, KREED-XL, with identical architecture to the flow model. Under an equal architecture, Stiefel FM obtains a higher success rate per number of function evaluations, which demonstrates that the improvement results from the new generative approach.
> >
> > **Ill-sentence.** Thank you for catching this.

---

> > > ### Comment · Reviewer_oa92 · 2024-11-22
> > > **Thank you for your response**
> > >
> > > I would like to thank the authors for their efforts and responses.
> > >
> > > I understand that the train/test split is applied, but my question was on OOD data. I suggest the authors to include the discussion of the OOD evaluation in the revised manuscript. If the method can generalize well due to the nature of the dataset, then I recommend highlighting this point.
> > >
> > > The authors do address most of my concerns. As such, I will raise my score.

---

### Official Review · Reviewer_f4fB · 2024-11-04

**Soundness:** 3
**Presentation:** 3
**Contribution:** 2
**Rating:** 6
**Confidence:** 3

**Summary:**

The paper proposes a generetive model approach for generating 3D molecular structures conditioned on the molecular formula and moments of inertia. The generative method is an extension of Riemannian flow matching with equivariant optimal transport on the Stiefel manifold.

**Strengths:**

The paper shows the effectiveness of the proposed approach through evaluations over two molecule benchmarks: QM9 and GEOM. The method achieved lower RMSD for the generated molecules with a lower number of function evaluations (NFE) compared to the Euclidean diffusion model.

**Weaknesses:**

Limited model comparisons: The paper only compares its results against one method in the literature, where it might consider other works on Riemannian generative models.

**Questions:**

Does the Stiefel manifold have any constraints? How does it handle the increased complexity of larger molecules and the number of conformers?

---

> ### Author Response · Authors · 2024-11-22
>
> Thank you for your comments on the paper. We appreciate your recognition of the effectiveness of our evaluation for our proposed method and the fact that we achieve better RMSD with a lower number of function evaluations. We address your comments below.
>
> **Limited baselines.** See global response.
>
> **Stiefel manifold constraints.** The Stiefel manifold *imposes* the constraints of moments of inertia. The only other constraint is that the molecule must be energetically stable, which should be learned by the model.
>
> **Large molecules.** The performance of Stiefel flow matching on large molecules is shown by its performance on GEOM. All conformers are used during training, but evaluation considers only the lowest energy conformer. This is because in a rotational spectroscopy measurement, the sample is cooled, and so the lowest-energy conformer is likely to have the highest population, which therefore shows the strongest signal in the spectrometer.
>
> [1] Hoogeboom, E., Satorras, V. G., Vignac, C., & Welling, M. (2022, June). Equivariant diffusion for molecule generation in 3d. In International conference on machine learning (pp. 8867-8887). PMLR.

---

### Official Review · Reviewer_Locv · 2024-11-04

**Soundness:** 3
**Presentation:** 4
**Contribution:** 4
**Rating:** 8
**Confidence:** 3

**Summary:**

This paper addresses the molecular structure elucidation problem using a generative approach. This task involves inferring a molecule's 3D structure from its molecular formula and moments of inertia. Traditional generative models conditioned on moments of inertia fail to leverage the precision offered by rotational spectroscopy, which measures moments with high accuracy. The paper introduces Stiefel Flow Matching, a generative model that operates on the Stiefel manifold, where the set of point clouds with fixed moments of inertia is embedded. Empirical results demonstrate higher accuracy and efficiency in generating 3D molecular structures compared to Euclidean diffusion models.

**Strengths:**

- As far as I can tell, the approach used in the paper of embedding the molecular structure elucidation task in the Stiefel manifold is original. This approach leverages manifold geometry to respect exact moment constraints, improving over traditional Euclidean diffusion models.
- In general, the paper is written and formatted very well. I appreciate the consistent and well-written formalisms and figures, making the otherwise quite abstract paper well-readable. Moreover, the formalisms support the methodology of the work well.
- The results show some performance improvements, particularly in maintaining moment constraints and reducing sampling cost.
- This work has potential applications in chemistry, pharmacology, and materials science, where precise molecular structures are essential. By leveraging constraints from rotational spectroscopy, this model could significantly improve the accuracy of 3D structure predictions for unknown molecules.

**Weaknesses:**

- Most obviously, for larger datasets like GEOM, the model struggles with validity and stability of generated structures, indicating possible underfitting issues. I think some more concrete suggestions on how to improve these issues would be greatly beneficial.
- Some connections to recent approaches to molecular generation with discrete flow matching are missing in the paper. Even though these problems consider a slightly different task as they operate on discrete domains, some intuition of why the authors' approach does not involve discrete dynamics/generation at all would be useful, as it intuitively would be beneficial for this task.
- As far as I understand it, some of the alignment and validation steps rely on heuristic checks. Some reflection on how task-specific these are, or some intuition behind the thereby introduced bias would be useful.
- While the authors provide some baseline comparisons, they are somewhat limited and some discussion on how Stiefel Flow Matching specifically outperforms or complements existing diffusion models on continuous or Riemannian spaces could benefit the paper. Moreover, adding more experimental results would aid in this too.
- The paper is longer than 9 pages.

**Questions:**

- The model seems to underfit the GEOM dataset, with low stability and validity rates. Could techniques from recent flow matching techniques or regularization methods improve performance? Alternatively, could a hybrid approach that combines Stiefel Flow Matching and discrete approaches maybe be beneficial here?
- While optimal transport reduces trajectory length, it appears to slightly reduce success rates. Would a more flexible trade-off strategy that selectively applies optimal transport based on the target molecule’s complexity yield better outcomes? Some comparisons here would be nice. On a similar note, the paper embeds the structure space on the Stiefel manifold, which assumes exact moment constraints. Have the authors considered using approximate embeddings on simpler manifolds for faster computation at the expense of slight constraint violations?
- Do I understand correctly the canonical metric on the Stiefel manifold is used? Given recent developments in Riemannian FM, did the authors explore alternative Riemannian metrics?

---

> ### Author Response · Authors · 2024-11-22
>
> Thank you for your thoughtful review of the paper. We appreciate your support of the work for its originality, readable presentation, and practical utility. We address your comments below:
>
> **Underfitting GEOM, concrete directions for improvement.** The difficulty of Stiefel FM may be attributed to pathologies in Riemannian flow matching with the geodesic distance on compact manifolds:
> 1. The velocity field under the geodesic distance suffers from discontinuity at the cut locus, as argued in [1].
> 2. The time-dependent probability density suffers from a shrinking support, which has been discussed for cases like the simplex [2] or $SO(3)$ (Appendix G.2 of [3]).
>
> Other works empirically report degraded results of Riemannian flow matching versus Riemannian diffusion on the torus [1,4]. These pathologies motivate the development of other probability paths for Stiefel flow matching, such as Stiefel diffusion, or flows which asymptotically land on the Stiefel manifold [6], both of which are discussed for future work.
>
> At sample time, some potential improvements for flow matching are to use corrector sampling [5], or to enhance the flow with a jump process [3]. At training time, we have tried other flow matching techniques, such as a logitnormal timestep schedule [7] and stochastic flow matching [8], but with limited improvement. Minibatch optimal transport [9] is technically challenging to apply here because each example has a variable number of atoms.
>
> **Connections with discrete flow matching.** In our setting, the only degrees of freedom are the continuous atom positions, which does not include the discrete atomic identities because we assume a fixed molecular formula. In future work where molecular formula is unknown, the jump processes of generator matching [3] could enable a multimodal flow which simultaneously varies continuous atom positions and discrete atom types. This process would jump between Stiefel manifolds corresponding to different molecular formulae. Importantly, jump processes would not require initial knowledge of the number of atoms.
>
> **Heuristic alignment and validation.** We discuss each method of alignment and validation.
> - The 3D alignment used in evaluating RMSD does not use a heuristic: it finds the node-permutation and reflection which yields the smallest least-squares Euclidean deviation. This is done by solving 8 linear assignment problems, one for each reflection, and taking the minimum.
> - The alignment of noise and data samples for optimal transport minimizes Stiefel distance using greedy randomized search. We cannot rely on a linear assignment solver since we must minimize the nonlinear Stiefel distance. We suspect that solving this assignment problem is more difficult than the quadratic assignment problem, which is NP-hard already, and leave full validation of this hypothesis to future work. We have observed that decreasing the number of iterations of the heuristic can yield suboptimal paths, but can still provide useful learning signal. This is in line with other works where optimal transport approximated using mini-batches can still provide valuable learning signal [9].
> - The validity metric uses rdkit's `rdDetermineConnectivity`, which assigns bonds based on a lookup of bond lengths and valency considerations.
>
> **Limited baselines.** See global response.
>
> **Longer than 9 pages.** The maximum page length is 10 according to the [call for papers](https://iclr.cc/Conferences/2025/CallForPapers):
> > the main text must be between 6 and 10 pages (inclusive). ... We encourage authors to be crisp in their writing by submitting papers with 9 pages of main text. We recommend that authors only use the longer page limit in order to include larger and more detailed figures. However, authors are free to use the pages as they wish, as long as they obey the page limits.

---

> > ### Author Response · Authors · 2024-11-22
> >
> > **Optimal transport appears to hurt success.** It is not necessarily true that optimal transport reduces success rate - in fact it improves success rate on GEOM. What is true for both datasets is that the sample generation paths that land on the true structure often are longer than the sample generation paths of incorrect structures. We show histograms in Appendix Figure 7 to this end. We are unsure of the reason for this correlation, but one explanation is that initial points are sampled anywhere uniformly on the manifold, whereas for success they must travel to the single true structure. In contrast, there are many incorrect structures all over the manifold, which may end up on average closer to random initial points.
> >
> > Regarding adaptive tradeoff, we could not find a meaningful difference that discriminates between success cases of Stiefel FM and Stiefel FM-OT, and we tried number of atoms, number of distinct atom types, total molecular weight. We note it is easy and appropriate to simply train and sample from multiple models, as the real-world scenario assumes large computational resources.
> >
> > For relaxing constraints, see above comment on flows which land on the Stiefel manifold.
> >
> > **Canonical and alternative metrics.** A one-parameter family of metrics on the Stiefel manifold exists [10], which includes the canonical metric as a special case. We use the canonical metric exclusively because the fast algebraic algorithm to compute the logarithm [11] only supports the canonical metric. Recent algorithms [12] extend the algebraic approach to the family of metrics, but we still find the canonical metric to be the fastest.
> >
> > All discussion has been included in the revised paper.
> >
> > [1] Lou, A., Xu, M., Farris, A., & Ermon, S. (2023). Scaling Riemannian diffusion models. Advances in Neural Information Processing Systems, 36, 80291-80305.
> >
> > [2] Stark, H., Jing, B., Wang, C., Corso, G., Berger, B., Barzilay, R., & Jaakkola, T. (2024). Dirichlet flow matching with applications to dna sequence design. arXiv preprint arXiv:2402.05841.
> >
> > [3] Holderrieth, P., Havasi, M., Yim, J., Shaul, N., Gat, I., Jaakkola, T., ... & Lipman, Y. (2024). Generator Matching: Generative modeling with arbitrary Markov processes. arXiv preprint arXiv:2410.20587.
> >
> > [4] Zhu, Y., Chen, T., Kong, L., Theodorou, E. A., & Tao, M. (2024). Trivialized Momentum Facilitates Diffusion Generative Modeling on Lie Groups. arXiv preprint arXiv:2405.16381.
> >
> > [5] Gat, I., Remez, T., Shaul, N., Kreuk, F., Chen, R. T., Synnaeve, G., ... & Lipman, Y. (2024). Discrete flow matching. arXiv preprint arXiv:2407.15595.
> >
> > [6] Gao, B., Vary, S., Ablin, P., & Absil, P. A. (2022). Optimization flows landing on the Stiefel manifold. IFAC-PapersOnLine, 55(30), 25-30.
> >
> > [7] Esser, P., Kulal, S., Blattmann, A., Entezari, R., Müller, J., Saini, H., ... & Rombach, R. (2024, March). Scaling rectified flow transformers for high-resolution image synthesis. In Forty-first International Conference on Machine Learning.
> >
> > [8] Bose, A. J., Akhound-Sadegh, T., Huguet, G., Fatras, K., Rector-Brooks, J., Liu, C. H., ... & Tong, A. (2023). Se (3)-stochastic flow matching for protein backbone generation. arXiv preprint arXiv:2310.02391.
> >
> > [9] Tong, A., Fatras, K., Malkin, N., Huguet, G., Zhang, Y., Rector-Brooks, J., ... & Bengio, Y. (2023). Improving and generalizing flow-based generative models with minibatch optimal transport. arXiv preprint arXiv:2302.00482.
> >
> > [10] Hüper, K., Markina, I., & Leite, F. S. (2021). A Lagrangian approach to extremal curves on Stiefel manifolds. AIMS.
> >
> > [11] Zimmermann, R., & Hüper, K. (2022). Computing the Riemannian logarithm on the Stiefel manifold: Metrics, methods, and performance. SIAM Journal on Matrix Analysis and Applications, 43(2), 953-980.
> >
> > [12] Mataigne, S., Zimmermann, R., & Miolane, N. (2024). An efficient algorithm for the Riemannian logarithm on the Stiefel manifold for a family of Riemannian metrics. arXiv preprint arXiv:2403.11730.

---

> > > ### Comment · Reviewer_Locv · 2024-11-24
> > >
> > > I want to thank the authors for their comments. To me, the contribution by the authors seems novel and insightful. I will raise my score.

---

### Author Response · Authors · 2024-11-22

We thank all reviewers for their thoughtful and constructive feedback. We are glad that reviewers Reviewers *Locv*, *oa92*, and *519h* recognize our application of the Stiefel manifold to molecular structure elucidation to be “original” and “the very first work”, and that they find our mathematical presentation of the Stiefel manifold and Stiefel geodesics to be “thorough”, “easy to follow”, and “consistent and well-written”. Reviewers *Locv*, *f4fB*, and *519h* also recognize that our contribution empirically improves both the accuracy and efficiency of moment-constrained structure elucidation, with practical “applications in chemistry, pharmacology, and materials science”.

**Empirical utility.** We first make a point that, despite tackling a heavily underconstrained problem, our results show that it is actually possible to take molecular formula and 3 moments of inertia, and then generate the all-atom 3D structure at 0.25 Å resolution, for:
- 27.4% of the test set of QM9 (3580/13033), when combining KREED-XL, Stiefel FM, and Stiefel FM-OT
- 7.9% of the test set of GEOM (2297/29203), when combining KREED-XL, Stiefel FM (filter), and Stiefel FM-OT (filter)

We now respond to common comments.

**Limited baselines.** Reviewers *Locv*, *f4fB*, and *519h* commented that Stiefel FM was compared to limited baselines. We would like to clarify a few items:
1. KREED-XL is a reflection-equivariant diffusion model that is a specialization of approaches like E(3)-equivariant diffusion [1], with differences being that KREED-XL does not change atom types, that it relaxes equivariance from E(3) to reflections, and has a more expressive architecture.
2. KREED-XL is different from KREED (Cheng et al.) because KREED assumes access to unsigned substitution coordinates as extra input, and they have different architectures.
3. KREED-XL-DPS is a significant baseline because Diffusion Posterior Sampling explicitly incorporates the closed-form formula of the moments in an extra guidance term while sampling the diffusion model.
4. We add an additional baseline, KREED-XL-proj, which simply takes all samples of KREED-XL and projects them onto the feasible manifold. The projection satisfies moment constraints exactly and does not change success rate, though it slightly reduces stability, since it has no effect on correct structures while distorting incorrect structures.

[1] Hoogeboom, E., Satorras, V. G., Vignac, C., & Welling, M. (2022, June). Equivariant diffusion for molecule generation in 3d. In International conference on machine learning (pp. 8867-8887). PMLR.

**Underfitting GEOM.** Reviewers *Locv* and *519h* commented that Stiefel FM on GEOM struggles with generating valid and stable structures. We point out that these validity and stability metrics are *averaged over all generated samples*, so validity and stability appear worse because when Stiefel FM fails, it tends to output a very bad structure. But, when Stiefel FM *does* generate the correct structure, it is valid and stable, which makes validity and stability a useful filtering mechanism for Stiefel FM, but not for KREED-XL. We note the greater efficiency (in terms of success rate per NFE) of Stiefel FM-OT for elucidating structure *despite* underfitting the dataset.

---

### Meta-Review · Area_Chair_QJD6 · 2024-12-27

**Metareview:**

The submission studies moment-constrained molecular structure identification, in which we are given the chemical formula of a molecule and the moments of its 3d structure. The goal is to sample from a distribution over structures which are both chemically realistic and agree with the observed moments. In previous work, this problem had been investigated using soft moment constraints. Here, the paper shows how to exactly enforce moment constraints, observing that these constrains can be converted into a manifold constraint in which allowable structures reside on a geodesic submanifold of the stiefel manifold. The paper develops generative models on this Stiefel manifold using flow matching and optimal transport.

The paper provides a novel approach to moment constrained structure generation. In particular, it provides a novel formulation of this problem as generation of a submanifold of the Stiefel manifold. Experiments demonstrate that the proposed method generates a larger fraction of correct structure compared to existing baselines. While there are some limitations to the method’s performance (this is a highly underdetermined inverse problem; all current methods struggle with the majority of instances). After discussion, reviewers converged to a uniform recommendation to accept, praising the paper’s novel and rigorous formulation and its experimental results.

**Additional Comments On Reviewer Discussion:**

The initial evaluation was mixed. On the positive side, reviewers noted that the paper develops a novel connection between generating structures with moment constraints and generative modeling on the Stiefel manifold. Reviewers found the paper to be very clearly written. The main points of discussion include:

- Performance on larger datasets such as GEOM [Locv,519h]. As noted by the author response, although the method struggles to generate valid and stable structures for some instances, it exhibits a higher success rate compared to existing methods.
- Optimal transport does not necessarily improve performance [Locv]. As noted by the authors, this is dataset dependent.
- Comparison to other models [Locv,f4fB,519h]: the author response added additional comparisons showing reduced computational cost compared to baselines.

Reviewers found that the author response addressed their concerns, and converged to a uniform recommendation of acceptance.

---

### Decision · Program_Chairs · 2025-01-22

Accept (Poster)